 **eLIFE**

# Attenuation of AMPK signaling by ROQUIN promotes T follicular helper cell formation

**Roybel R Ramiscal[1]\*, Ian A Parish[1], Robert S Lee-Young[2], Jeffrey J Babon[3], Julianna Blagih[4], Alvin Pratama[1], Jaime Martin[1], Naomi Hawley[1], Jean Y Cappello[1], Pablo F Nieto[1], Julia I Ellyard[1], Nadia J Kershaw[3], Rebecca A Sweet[1], Christopher C Goodnow[1,5], Russell G Jones[4], Mark A Febbraio[2,6], Carola G Vinuesa[1†], Vicki Athanasopoulos[1\*†]**

[1]Department of Immunology and Infectious Disease, John Curtin School of Medical Research, Australian National University, Canberra, Australia; [2]Cellular and Molecular Metabolism Laboratory, Baker IDI Heart and Diabetes Institute, Melbourne, Australia; [3]Division of Structural Biology, Walter and Eliza Hall Institute of Medical Research, Melbourne, Australia; [4]Department of Physiology, Goodman Cancer Research Centre, McGill University, Montreal, Canada; [5]Immunology Division, Garvan Institute of Medical Research, Sydney, Australia; [6]Diabetes and Metabolism Division, Garvan Institute of Medical Research, Sydney, Australia

**\*For correspondence:** roy. ramiscal@anu.edu.au (RRR); vicki. athanasopoulos@anu.edu.au (VA)

[†]These authors contributed equally to this work

**Competing interests:** The authors declare that no competing interests exist.

**Abstract** T follicular helper cells (Tfh) are critical for the longevity and quality of antibody-mediated protection against infection. Yet few signaling pathways have been identified to be unique solely to Tfh development. ROQUIN is a post-transcriptional repressor of T cells, acting through its ROQ domain to destabilize mRNA targets important for Th1, Th17, and Tfh biology. Here, we report that ROQUIN has a paradoxical function on Tfh differentiation mediated by its RING domain: mice with a T cell-specific deletion of the ROQUIN RING domain have unchanged Th1, Th2, Th17, and Tregs during a T-dependent response but show a profoundly defective antigen-specific Tfh compartment. ROQUIN RING signaling directly antagonized the catalytic $\alpha$1 subunit of adenosine monophosphate-activated protein kinase (AMPK), a central stress-responsive regulator of cellular metabolism and mTOR signaling, which is known to facilitate T-dependent humoral immunity. We therefore unexpectedly uncover a ROQUIN–AMPK metabolic signaling nexus essential for selectively promoting Tfh responses.

## Introduction

High-affinity and long-lasting humoral immunity against infection requires controlled cross-talk between limiting CD4$^+$CXCR5$^{high}$PD1$^{high}$BCL6$^{high}$ T follicular helper (Tfh) cells and immunoglobulin-maturing germinal center (GC) B cells in secondary lymphoid tissues (**King et al., 2008**; **Victora and Nussenzweig, 2012**; **Nutt and Tarlinton, 2011**; **Ramiscal and Vinuesa, 2013**). As the GC largely consists of clonally diverse B cells, Tfh cells especially in narrow numbers are best at maintaining a selective pressure for B cell competition, favoring the survival of greater affinity antigen-responsive GC B cell clones (**Pratama and Vinuesa, 2014**; **Victora and Mesin, 2014**). Deregulation of Tfh cells can lead to faulty GC selection that may also seed the production of autoantibodies (**Weinstein et al., 2012**; **Vinuesa et al., 2005**; **Kim et al., 2015**; **Linterman et al., 2009**) and GC-derived malignancies such as follicular lymphoma (**Rawal et al., 2013**; **Klein and Dalla-Favera,**

**eLife digest** The immune system protects the body from invading microbes like bacteria and viruses. Upon recognizing the presence of these microbes, cells in the immune system are activated to destroy the foreign threat and clear it from the body.

A type of immune cell called T follicular helper cells (or Tfh for short) are formed during an infection and are essential for coordinating other immune cells to produce high-quality antibody proteins that attack the microbes. Without Tfh cells, life-long production of these protective antibodies is severely crippled, which can cause common variable immune deficiency and other serious immunodeficiency diseases. On the other hand, the body must also avoid generating excessive numbers of Tfh cells, which can lead to the production of antibodies that attack healthy cells of the body.

ROQUIN is a protein that inhibits the formation of Tfh cells and other types of active T cells. A region on the protein called the ROQ domain destabilizes particular molecules of ribonucleic acid (RNA) that are required for these specialist T cells to form and work properly. ROQUIN belongs to a large family of enzymes that have a so-called RING domain, which is a feature that enables these enzymes to attach tags onto specific target proteins to modify their activity or stability. However, it was not known whether the RING domain of ROQUIN was active.

Ramiscal et al. now address this question in mice. Unexpectedly, the experiments show that the RING domain is required to promote the formation of Tfh cells, but not other types of active T cells. This domain allows ROQUIN to repress an enzyme called AMPK, which normally blocks cell growth by regulating cell metabolism. The findings suggest that the different roles of the ROQ and RING domains allow ROQUIN to fine-tune the numbers of Tfh cells so that they remain within a safe range. In the future, these findings may aid the development of vaccines that are more efficient at generating protective Tfh cells to prevent infectious diseases.

*2008*). To date, the signals that exclusively govern Tfh cell differentiation over other T cell effector subsets remains poorly characterized.

ROQUIN (also called ROQUIN1; encoded by *Rc3h1*) acts to post-transcriptionally repress Tfh cells by binding effector T cell transcripts via its winged-helix ROQ domain (*Schuetz et al., 2014*; *Tan et al., 2014*; *Schlundt et al., 2014*) and recruiting proteins of the RNA decapping and deadenylation machinery (*Athanasopoulos et al., 2010*; *Glasmacher et al., 2010*; *Leppek et al., 2013*; *Pratama et al., 2013*; *Yu et al., 2007*; *Vogel et al., 2013*) as well as the endoribonuclease REGNASE-1 (*Jeltsch et al., 2014*). Some of its RNA targets include the Tfh-polarising *Icos* (*Glasmacher et al., 2010*) and *Il6* mRNA (*Jeltsch et al., 2014*) as well as *Ox40* (*Vogel et al., 2013*) and *Tnf* (*Pratama et al., 2013*) transcripts. In *sanroque* mice, an *Rc3h1* missense point mutation, encoding for a Met$^{199}$ to Arg substitution translates into a minor conformational shift in the RNA-binding ROQ domain (*Srivastava et al., 2015*) of ROQUIN and a loss of function in post-transcriptional repression. This leads to excessive Tfh growth and systemic autoimmunity (*Linterman et al., 2009*; *Vinuesa et al., 2005*). Complete ablation of ROQUIN results in unexplained perinatal lethality in C57BL/6 mice and selective deletion of ROQUIN in T cells does not lead to Tfh cell accumulation nor autoimmunity (*Bertossi et al., 2011*). The latter is at least in part explained by the existence of the closely related family member ROQUIN2 (encoded by *Rc3h2*), which has overlapping functions with ROQUIN (*Pratama et al., 2013*; *Vogel et al., 2013*). The ROQUIN$^{M199R}$ mutant protein has been proposed to act as a 'niche-filling' variant that has lost its RNA-regulating activity (*Pratama et al., 2013*) but can still localize to mRNA-regulating cytoplasmic granules to prevent the compensatory activity of ROQUIN2.

ROQUIN contains a conserved amino terminal RING finger with two conforming zinc-chelating sites (*Srivastava et al., 2015*), despite an atypical aspartate as its eighth zinc ligand synonymous to RBX1 (*Kamura et al., 1999*). This suggests ROQUIN may function as an E3 ubiquitin ligase (*Deshaies and Joazeiro, 2009*) but, to date, no such enzymatic activity of the ROQUIN RING domain has been demonstrated in mammals. *In vivo* attempts to delineate the cellular pathways regulated by ROQUIN are made challenging due to the existence of multiple protein domains in the protein (*Figure 1—figure supplement 1a*). The *Caenorhabditis elegans* ROQUIN ortholog, RLE-1,

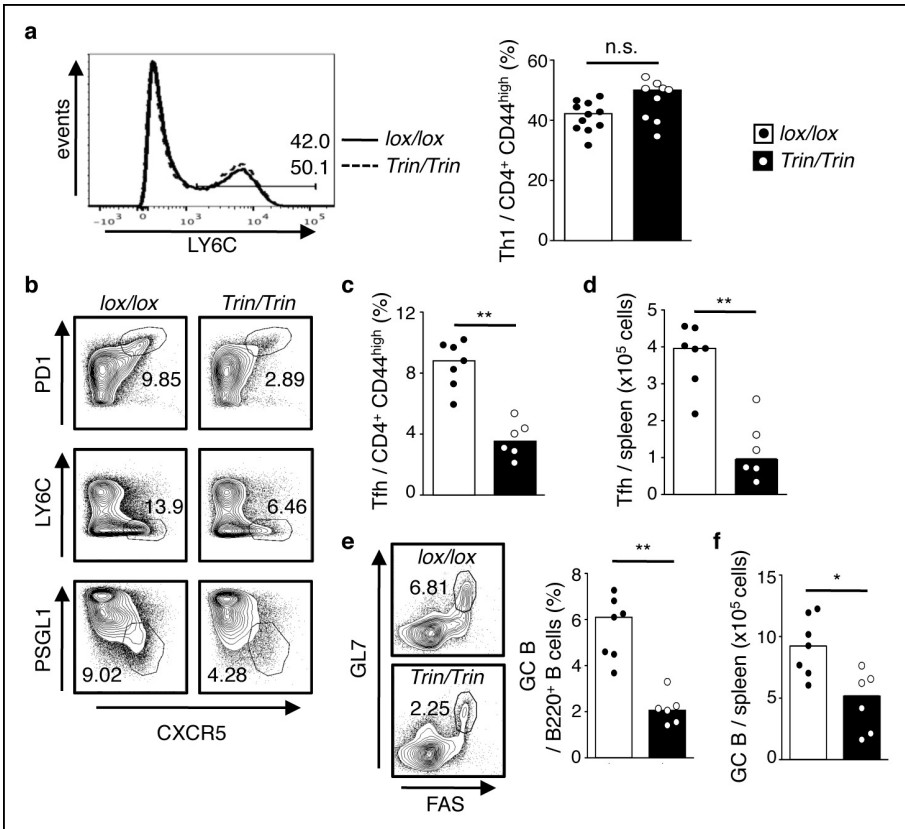

**Figure 1.** ROQUIN RING deletion in T cells preferentially controls Tfh cell formation. (a-f) Flow cytometric examination of mice d10 post-LCMV infection. (a) Proportion of LY6C$^+$ total Th1 cells from CD4$^+$CD44$^{high}$ T cells. (b) Identification of total Tfh cells pre-gated on CD4$^+$CD44$^{high}$ T cells. (c) Proportion of PD1$^{high}$CXCR5$^{high}$ Tfh cells from CD4$^+$CD44$^{high}$ T cells. (d) PD1$^{high}$CXCR5$^{high}$CD44$^{high}$ Tfh cell numbers from spleen. (e) Proportion and (f) cell count of GL7$^{high}$FAS$^{high}$ GC B cells in spleen. Data are pooled from three independent experiments (n = 2–3). Statistics were calculated by Student's t-test, n.s., not significant; *$p<0.05$; **$p<0.005$. Dot symbols, individual mice; columns, median.

The following figure supplements are available for Figure 1:

**Figure supplement 1.** Generation of mice with a ROQUIN RING deletion.

**Figure supplement 2.** Phenotype of mice with a T cell-specific ROQUIN RING deletion.

**Figure supplement 3.** Phenotype of SRBC-immunized mice with a T cell-specific ROQUIN RING deletion.

acts through its RING domain to ubiquitinate DAF-16, a pro-longevity forkhead box O (FOXO) transcription factor homolog (*Li et al., 2007*). We did not find any evidence for molecular binding between ROQUIN and the fruitfly or mammalian FOXO orthologs (*Drosophila melanogaster* FOXO and *Mus musculus* FOXO1 or FOXO3a; data not shown) and therefore set out to understand the role of ROQUIN RING signaling in CD4$^+$ T cell development and function by generating mice that selectively lack the ROQUIN RING zinc finger.

We previously demonstrated that ROQUIN RING-deleted T cells in mice 6 days after sheep red blood cell (SRBC) immunization can form normal early Tfh cell responses but fail to promote optimal GC B cell reactions (*Pratama et al., 2013*). Here, in mice that have developed robust Tfh-dependent GC responses toward SRBC or infected with lymphocytic choriomeningitis virus (LCMV), we identify a novel and unexpected role of the ROQUIN RING domain in selectively promoting mature antigen-specific Tfh cell responses while leaving unaffected the development of other CD4$^+$ effector T cell lineages. ROQUIN directly binds to and limits adenosine monophosphate-activated protein kinase (AMPK), a tumor suppressor and central regulator of T cell glucose uptake and glycolysis (*MacIver et al., 2011*). Our data indicate that loss of AMPK repression by deletion of the ROQUIN

RING domain promotes stress granule persistence. This in turn cripples mTOR activity, otherwise known to play a critical role in driving CD4$^+$ effector T cell expansion (*Delgoffe et al., 2009*; *2011*) and T-dependent antibody responses (*Keating et al., 2013*; *Zhang et al., 2011*; *Gigoux et al., 2014*; *De Bruyne et al., 2015*).

## Results

### The ROQUIN RING domain selectively controls Tfh cell formation

To examine the function of the ROQUIN RING domain *in vivo*, we generated two strains of C57BL/6 mice carrying either a germline deletion (designated *ringless*; '*rin*' allele) or a T cell conditional deletion (*Tringless*; '*Trin*' allele) of exon 2 in the *Rc3h1* gene, which encodes the translation START codon and RING finger domain of the ROQUIN protein (*Figure 1—figure supplement 1b, c* and *Pratama et al., 2013*). In these mice, skipping of exon 2 resulted in splicing of exon 1 to exon 3 yielding an alternative in-frame Kozak translation initiation site at Met$^{133}$ (*Figure 1—figure supplement 1d, e*). This predicted ROQUIN$^{133-1130}$ protein product specifically lacks the RING domain (*Figure 1—figure supplement 1f*). Mice homozygous for the *rin* allele were perinatally lethal (*Figure 1—figure supplement 1g–i*), precluding T cell studies in intact animals. In contrast, *Tringless* mice were viable and showed no severe variations in thymic development and output of CD4 single positive T cells (*Figure 1—figure supplement 2a–e*). There were also no major changes in Th1 cell differentiation in *Tringless* mice infected with LCMV (*Figure 1a*), which predominantly yields LY6C$^{high}$ Th1 and LY6C$^{low}$ Tfh virus-specific effector cells (*Hale et al., 2013*; *Marshall et al., 2011*). In *Tringless* animals immunized with SRBCs, the formation of Th1, Th2, Th17, and regulatory T cells also remained largely unperturbed (*Figure 1—figure supplement 2f, g*). This was mirrored *in vitro* with *Tringless* CD4$^+$ naive T cells activated under Th1, Th2, Th17, or induced Treg (iTreg) polarizing conditions (*Figure 1—figure supplement 2h*) displaying maximal expression of intracellular TBET, GATA3, RORγT, and FOXP3 comparable to floxed wild-type T cell cultures (*Figure 1—figure supplement 2i*). Surprisingly in *Tringless* mice, there was an overall defective Tfh cell primary response to LCMV infection (*Figure 1b–d*) and to SBRC immunization (*Figure 1—figure supplement 3a*). ROQUIN RING-deficient T cells were also inefficient in supporting GC formation (*Figure 1e, f* and *Figure 1—figure supplement 3b*), which was associated with reduced IL-21 production (*Figure 2a*), a Tfh signature cytokine vital in supporting GC reactions (*Liu and King, 2013*).

By stimulating splenocytes *ex vivo* with GP$_{61-80}$ peptide to identify virus-responsive IFNγ-producing Th1 cells (*Figure 2b*) and by examining splenic LYC6$^{high}$ Th1 cells amongst GP$_{66-77}^+$ tetramer stained T cells (*Figure 2c*), we verified that ROQUIN RING loss did not disrupt protective Th1 responses but caused a severe abrogation of virus-specific Tfh cells during LCMV infection (*Figure 2d–f*). Virus-specific T cells also showed significantly reduced expression of BCL6 (*Figure 2g*), an indispensible nuclear factor for Tfh cell terminal differentiation (*Liu et al., 2013*). Furthermore, we found an increased frequency of FOXP3$^+$ T follicular regulatory (Tfr) cells within the total Tfh pool (*Figure 2h*) despite these Tfr cells not expressing a GP$_{66-77}$ virus-specific T cell antigen receptor (TCR; *Figure 2i*). Nonetheless, as Tfr cells are negative regulators of GC reactions (*Ramiscal and Vinuesa, 2013*), their abundance may indicate augmented suppression of Tfh cells and long-term B cell responses.

### ROQUIN undergoes RING-dependent autoubiquitination and directly limits AMPK activity

We next sought to determine the molecular basis for the ROQUIN RING domain as a determinant in protective Tfh cell responses. Several lines of evidence implicated an involvement of ROQUIN in the negative regulation of AMPK signaling: *Rc3h1$^{ringless}$* fetuses displayed skeletal muscle atrophy of the thoracic diaphragm (*Figure 1—figure supplement 1j*), which is a characteristic phenotype of mice with overactive AMPK (*Sanchez et al., 2012*) and pointed to perinatal respiratory failure as the cause of the lethality. Also, AMPK over-expression in nematode worms has been shown to extend lifespan (*Mair et al., 2011*), an observation consistent with the phenotype of worms lacking the ROQUIN ortholog RLE-1 (*Li et al., 2007*). Since the AMPKα1 catalytic subunit is expressed in T cells and responds to TCR activation (*Tamas et al., 2006*), we tested the possibility of ROQUIN directly binding to this subunit of AMPK (encoded by *Prkaa1*). Upon ectopic expression in HEK293T cells,

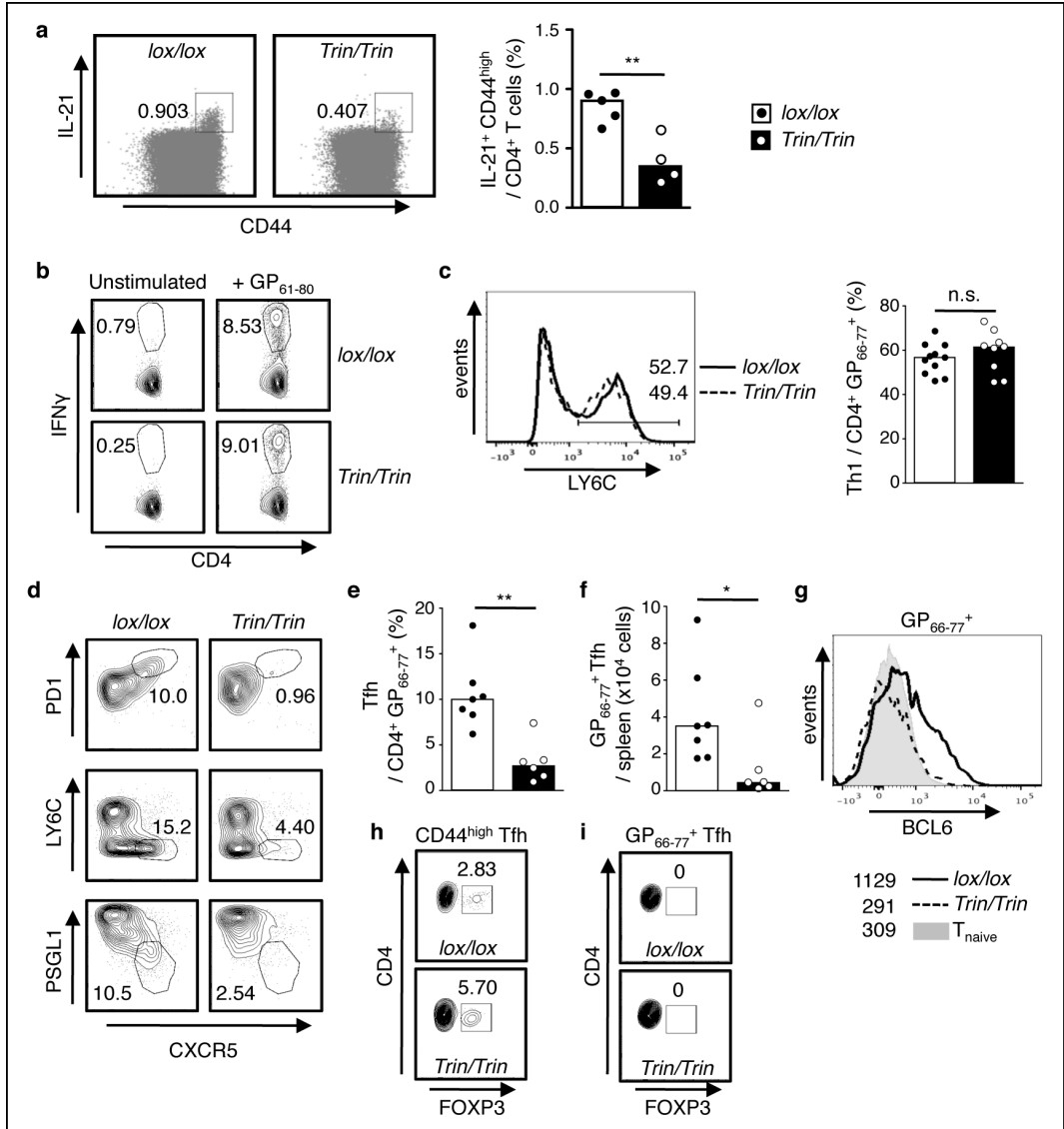

**Figure 2.** Functional competency of ROQUIN RING deleted Tfh cell responses. (**a**) Flow cytometric analysis of mice 8d after sheep red blood cell (SRBC) immunization showing the proportion of IL-21$^+$CD44$^{high}$ effectors from total CD4$^+$ T cells in the spleen. Data are representative of two independent experiments. (**b-i**) Flow cytometric examination of mice d10 post-lymphocytic choriomeningitis virus (LCMV) infection. (**b**) Proportion of IFN$\gamma^+$ Th1 cells gated from total CD4$^+$ T cells after GP$_{61-80}$ peptide stimulation *ex vivo*. (**c**) Proportion of LY6C$^+$ Th1 cells from virus-specific CD4$^+$GP$_{66-77}^+$ T cells. (**d**) Identification of virus-specific Tfh cells pre-gated on CD4$^+$GP$_{66-77}^+$ T cells. (**e**) Proportion of PD1$^{high}$CXCR5$^{high}$ Tfh cells from virus-specific CD4$^+$GP$_{66-77}^+$ T cells. (**f**) Virus-specific CD4$^+$PD1$^{high}$CXCR5$^{high}$GP$_{66-77}^+$ Tfh cell numbers in spleen. (**g**) Representative histograms of BCL6 expression in virus-specific CD4$^+$GP$_{66-77}^+$ T cells. Values included show median MFI for each genotype. (**h**) Proportion of FOXP3$^+$ Tfr cells within the total CD4$^+$CD44$^{high}$PD1$^{high}$CXCR5$^{high}$ Tfh gate. (**i**) Proportion of FOXP3$^+$ Tfr cells within the virus-specific CD4$^+$GP$_{66-77}^+$PD1$^{high}$CXCR5$^{high}$ Tfh gate. Data are pooled from three independent experiments (n = 2–3). Statistics were calculated by Student's t-test, n.s., not significant; *$p<0.05$; **$p<0.005$; Dot symbols, individual mice; columns, median.

ROQUIN colocalized with AMPKα1 diffusely or in fine cytoplasmic speckles in resting cells and within larger cytoplasmic granules upon induction of oxidative stress (*Figure 3a*). We also observed colocalization of endogenous AMPKα1 within ROQUIN$^+$ cytoplasmic granules in arsenite-treated primary C57BL/6 mouse embryonic fibroblasts (MEFs) (*Figure 3b*) with the use of an AMPKα1-specific antibody displaying no cross-reactivity toward the AMPKα2 subunit when ectopically expressed in HEK293T cells (*Figure 3—figure supplement 1a*). Unlike the AMPKα1 subunit, ectopically expressed AMPK β and γ regulatory subunits did not associate with ROQUIN$^+$ cytoplasmic granules, although AMPKγ2 and AMPKγ3 exhibited generally diffuse cytoplasmic distribution (*Figure 3—*

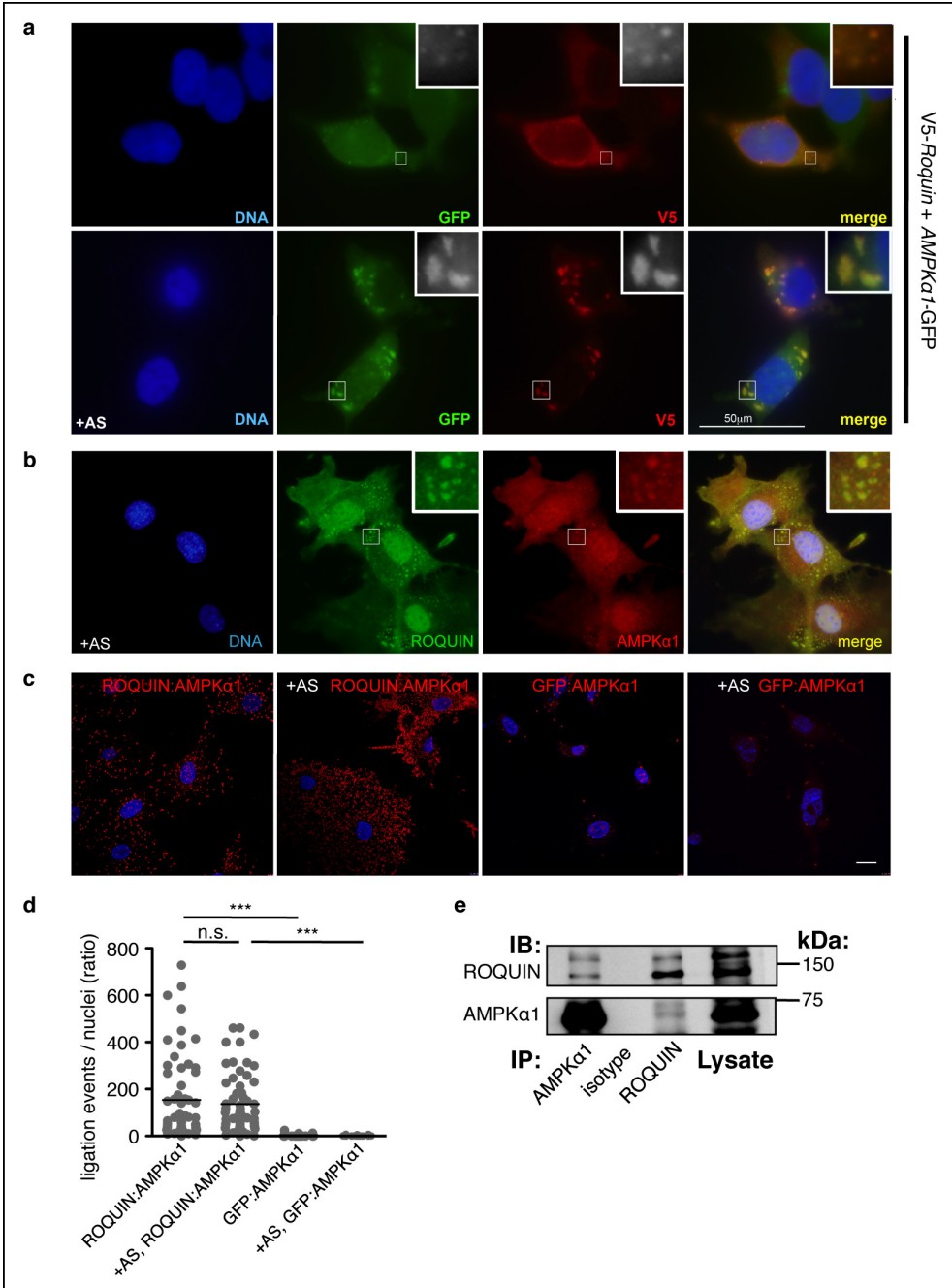

**Figure 3.** ROQUIN preferentially colocalizes and binds with the $\alpha 1$ subunit of AMPK. (**a**) Colocalization of V5-ROQUIN and AMPKα1-GFP ectopically expressed in resting (top) and 1 mM arsenite (AS)-treated (bottom) HEK293T cells. Representative of three independent experiments. (**b**) Colocalization of endogenous ROQUIN and AMPKα1 in primary (mouse embryonic fibroblasts) MEFs post-arsenite (AS) treatment. Representative of three independent experiments. (**c**) Proximity ligation assays (PLAs) performed on primary C57BL/6 MEFs showing interactions between endogenously expressed ROQUIN and AMPKα1 in resting cells (ROQUIN:AMPKα1) and in cells stressed with 1 mM arsenite (+AS, ROQUIN:AMPKα1). Negative control PLAs (GFP:AMPKα1) detecting non-expressed GFP and endogenous AMPKα1 background are also displayed. Blue, DAPI stained nuclei; Red, ligation events, Scale bar, 20 μm. Representative of three independent experiments. (**d**) Quantitative analysis of PLAs showing mean ligation events per cell (nucleus) for each field of view on a confocal microscope. Individual dots represent a single field of view; bar per column represents the sample mean. Statistics were calculated by one-way ANOVA with Bonferroni's multiple comparisons test after log transformation of ratio values, n.s., not significant;
*Figure 3. continued on next page*

*Figure 3. Continued*

***$p<0.0005$. (**e**) Reciprocal coimmunoprecipitation of ROQUIN and AMPKα1 endogenously expressed in EL4 cells. IB, immunoblot; IP, immunoprecipitated.
The following figure supplements are available for Figure 3:

**Figure supplement 1.** Association of AMPK subunits with ROQUIN.

*figure supplement 1b*). We next determined if ROQUIN and AMPKα1 interacted by conducting *in situ* proximity ligation assays (PLAs) on primary C57BL/6 MEFs. Compared to control PLAs accounting for false interactions between endogenous AMPKα1 and non-expressed green fluorescent protein (GFP) detected by optimized anti-GFP immunostaining (*Figure 3—figure supplement 1c*), we found that endogenously expressed ROQUIN and AMPKα1 proteins localized with very close molecular proximity in both resting and arsenite-stressed cells (*Figure 3c, d*) at a frequency 15-fold higher or more than weak PLA interactions previously observed between ROQUIN and AGO2 (*Srivastava et al., 2015*). Moreover, we were able to coimmunoprecipitate ROQUIN and AMPKα1 when over-expressed in HEK293T cells (*Figure 3—figure supplement 1d*) or expressed endogenously in the mouse T lymphoblast line EL4 cells (*Figure 3e*). Together with the PLAs, this indicated that ROQUIN bound specifically with the α1 subunit of AMPK and that under physiological conditions, the two proteins could form a stable complex.

To determine the functional consequence of a ROQUIN–AMPKα1 interaction, we measured AMPK activity in *Tringless* and wild-type T cells. In contrast to wild-type cells, phosphorylation of the AMPK target, acetyl CoA carboxylase (ACC) in ROQUIN RING-deficient CD4[+] T cells was increased, demonstrating constitutively active AMPK activity *in vitro* (*Figure 4a*) and *in vivo* (*Figure 4b*). Thus, ROQUIN acts through its RING domain to directly negatively regulate AMPKα1 activity in T cells.

Given the important role of RING domains in driving protein substrate ubiquitination (*Deshaies and Joazeiro, 2009*), we next tested if the regulation of AMPK activity by ROQUIN was a result of RING-mediated AMPK ubiquitination. Absence of the ROQUIN RING domain did not alter AMPK ubiquitination (data not shown). However, monoubiquitination of endogenous ROQUIN in EL4 cells was detected (*Figure 4c*). To determine if ROQUIN monoubiquitination was dependent on the 14.7 kDa RING finger deleted in ROQUIN RING deficient mice (*Figure 4d*), we tested if ROQUIN could undergo automonoubiquitination *in vitro* and in a cell-based ubiquitin assay. By Coomassie staining PAGE-separated peptides of *in vitro* ubiquitination reactions, we detected a single protein band having higher molecular weight relative to ROQUIN peptide that formed in the presence of wild-type ROQUIN[1-484] and ubiquitin (*Figure 4e*). This slowly migrating band, consistent with monoubiquitin attachment, formed at severely delayed times in the absence of the RING zinc finger. A complete absence of this higher molecular weight ROQUIN peptide modification was observed with *in vitro* reactions lacking ubiquitin protein. We also performed ubiquitination assays in transfected HEK293T cells and detected ubiquitin-conjugated ROQUIN by immunoprecipitation when full-length ROQUIN was over-expressed but not with expression of the ROQUIN[133-1130] variant recapitulating the specific RING deletion borne by *Tringless* T cells (*Figure 4f*). Together, our data show that the ROQUIN RING domain can facilitate automonoubiquitination independent of residues carboxy terminal to Asp[484].

We next investigated the mechanism by which ROQUIN RING activity limits AMPK signaling. Analogous to RAPTOR inactivation within stress granules (*Thedieck et al., 2013*; *Wippich et al., 2013*), we hypothesized that ROQUIN localization and its ability to bind AMPK within stress granules was key to AMPK repression. We have previously shown that ROQUIN[133-1130] lacking the RING domain did not coalesce with eIF3[+] stress granules (*Pratama et al., 2013*). To exclude the possibility that this mislocalization of RING-deficient ROQUIN was a product of over-active AMPK feedback, we investigated if AMPK hyperactivity prevented ROQUIN localizing to stress granules. Full length ROQUIN still colocalized with eIF3[+] stress granules in the presence of the AMPK agonist, AICAR (*Figure 5a*), which alone was ineffective at inducing stress granule formation (data not shown). This indicates that ROQUIN[133-1130] mislocalization is a direct consequence of an intrinsic lack of the RING domain. To confirm that stress granule exclusion was not a secondary effect of a structurally unstable ROQUIN[1-132] deletion but rather a consequence of the loss of RING-mediated E3 ligase activity, a

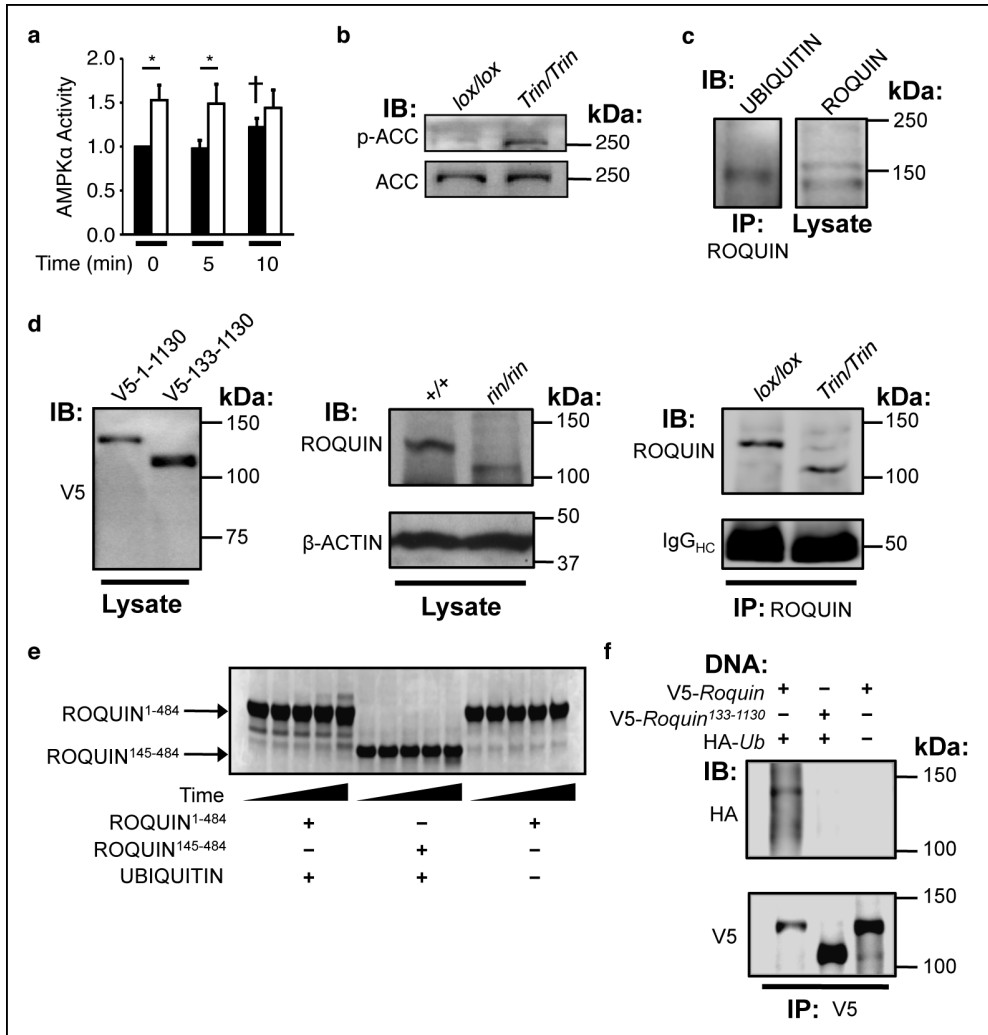

**Figure 4.** The ROQUIN RING finger is required for autoubiquitination and negative regulation of AMPK. (a) *In vitro* kinase assay of AMPKα in isolated CD4[+] T cells during an anti-CD3 and -CD28 activation time-course. Data are pooled from two independent experiments and normalized to unstimulated wild-type (n = 5). Black columns, *floxed* wild-type; white columns, *Tringless*. Statistics were calculated by Student's t-test, *$p<0.05$. †$p$ <0.05 for wild-type at 10 min vs. wild-type at 0 and 5 min. columns, mean; error bars, s.e.m. (b) Phospho-blot of endogenous ACC Ser[79] in resting CD4[+] T cells. Representative of three independent experiments. IB, Immunoblot. (c) Ubiquitin immunobDot of endogenous ROQUIN immunoprecipated from EL4 cells (d) Immunoblot of V5-tagged ROQUIN[1-1130] and ROQUIN[133-1130] in transfected HEK293T cells (left), endogenous ROQUIN in *ringless* primary MEFs (center), immunoprecipitated ROQUIN in *Tringless* thymocytes (right). (e) *In vitro* autoubiquitination assay for ROQUIN wild-type peptide (residues 1–484) and RING finger deleted peptide (residues 145–484). Five consecutive lanes show the extent of ROQUIN autoubiquitination of the same *in vitro* reaction at 0, 1, 2, 4, and 16 h. (f) Cellular ubiquitination assay for full length V5-ROQUIN and RING-deleted V5-ROQUIN[133-1130] ectopically expressed in HEK293T cells with HA-Ub. Data are representative of three independent experiments. IB, immunoblot; IP, immunoprecipitated.

loss-of-function mutation of the first zinc-coordinating cysteine of the RING domain (Cys[14]Ala; *Figure 5b*) that typically abolishes E3 ligase activity of related RING-containing enzymes (*Fang et al., 2001*; *2000*) was introduced into HEK293T cells. Although ROQUIN[C14A] ectopic expression could facilitate *de novo* stress granule induction in the absence of arsenite treatment comparable to cells transfected with wild-type ROQUIN (*Athanasopoulos et al., 2010*), we found that in response to arsenite exposure, ROQUIN[C14A] localization to eIF3[+] stress granules was significantly impaired (*Figure 5c*). A deleted RING domain did not abrogate ROQUIN[133-1130]-AMPKα1 colocalization; the two proteins were detected in small aggregates most likely outside of stress granules (*Figure 5d*). This was consistent with RING deficient ROQUIN[133-1130] protein still capable of directly binding AMPKα1 (*Figure 5e*). Together these findings indicate that ROQUIN RING signaling does not play a role in AMPK recruitment to ROQUIN but rather directs negative regulation of

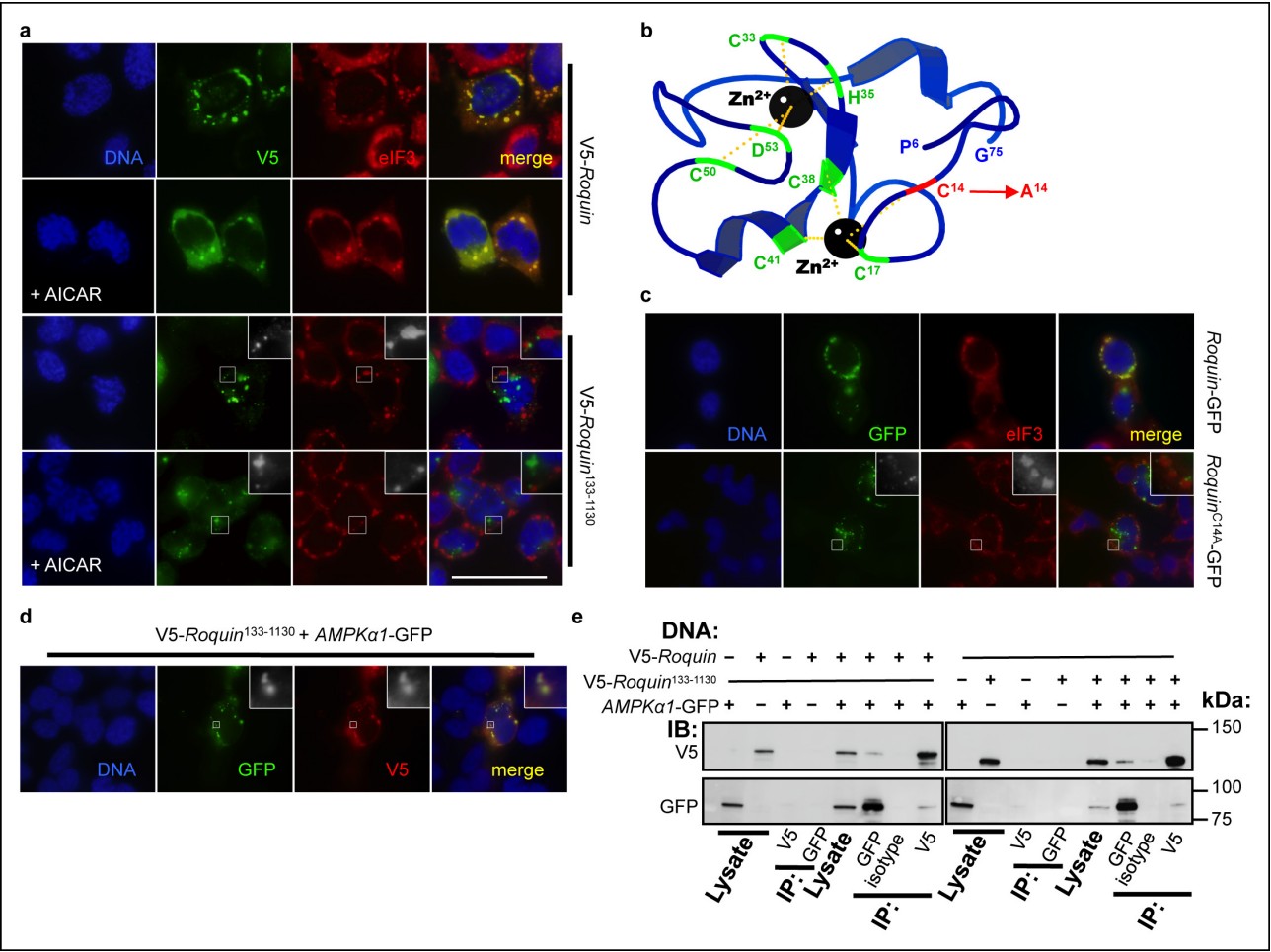

**Figure 5.** ROQUIN RING activity controls its localization to stress granules. (**a**) Colocalization of over-expressed full length V5-tagged ROQUIN or ROQUIN[133-1130] with endogenous eIF3 in HEK293T cells stressed with 1 mM arsenite (AS) for 1 hr with or without 2 mM AICAR. Scale bar, 50 μm. (**b**) Crystal structure of ROQUIN peptide showing amino terminal residues 6 to 75 incorporating the RING domain. Black, zinc cation; green, zinc-coordinating residue, red, zinc-coordinating Cys[14] targeted for mutagenesis; yellow, zinc-chelating interaction. Data are based on structural coordinates we had previously determined (*Srivastava et al., 2015*) and deposited in the Protein Data Bank, accession code 4TXA. (**c**) Colocalization of over-expressed full length GFP-tagged ROQUIN or ROQUIN[C14A] mutant with endogenous eIF3 in HEK293T cells stressed with 1 mM arsenite for 1 hr. (**d**) Colocalization of ROQUIN[133-1130] with AMPKα1 when over-expressed in HEK293T cells immediately after 1 mM arsenite exposure for 1 hr. (**e**) Reciprocal coimmunoprecipitation of full length ROQUIN or ROQUIN[133-1130] and AMPKα1 over-expressed in HEK293T cells. IB, immunoblot; IP, immunoprecipitated.

AMPKα1 through sequestration into stress granules following ROQUIN–AMPKα1 complex formation.

## ROQUIN RING loss results in stress granule longevity and dampened mTOR

One possible downstream effector of ROQUIN–AMPK in Tfh cells is the mechanistic Target of Rapamycin (mTOR), a nutrient sensing kinase and modulator of cellular metabolism. AMPK activity directly suppresses mTORC1 signaling (*Gwinn et al., 2008*; *Inoki et al., 2003*), and deletion of AMPKα1 increases mTORC1 signaling in T cells (*MacIver et al., 2011*). Although the role of mTOR in promoting effector CD4[+] and CD8[+] T cell responses is well documented (*Araki et al., 2011*; *Chi, 2012*), mTOR signaling in Tfh cell formation, and therefore antibody responses, is incompletely understood. In this respect, we assessed mTOR function in *Tringless* CD4[+] T cells in response to TCR and CD28 stimulation. In CD4[+] T cells, we found a reduction in phosphorylated ribosomal S6 in the absence of ROQUIN RING, indicating diminished mTORC1 function (*Figure 6a*). This effect was

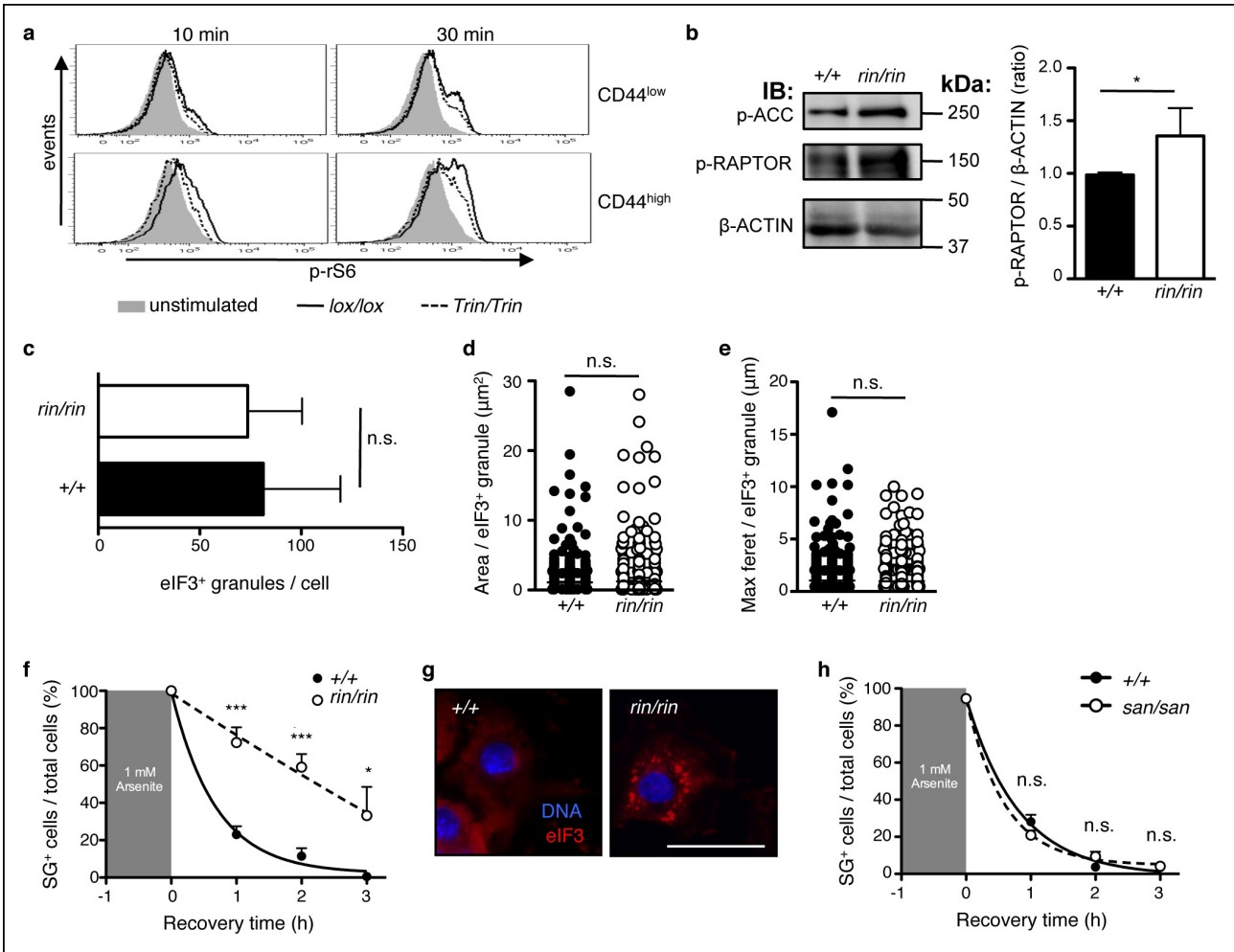

**Figure 6.** ROQUIN RING signaling regulates stress granule responses to promote mTOR. (a) Flow cytometric analysis of phospho-rS6 Ser[235/236] in CD44[low] or CD44[high] anti-CD3 and anti-CD28 stimulated CD4[+] T cells (n = 4–6). (b) Phosphoblot of ACC Ser[70] and RAPTOR Ser[792] in primary mouse embryonic fibroblasts (MEFs) recovered in complete DMEM for 3 hr after 1 hr of 1 mM arsenite treatment (left). Quantitative ratios of phosphorylated RAPTOR to β-ACTIN input based in phosphoblot MFI readings (right). IB, immunoblot. (c-e) Analysis of stress granule induction in primary MEFs analyzed by fluorescence microscopy after 1 hr of 1 mM arsenite stress treatment showing counts of eIF3[+] granules per cell (c), and size of individual eIF3[+] granules in freshly arsenite-stressed primary MEFs based on area (d) and maximum feret (e). (f) Proportion of recovering primary MEFs exhibiting cytoplasmic eIF3[+] stress granules (SG) after arsenite-mediated stress (n >30 per time point, with each n replicate representing a single field of view displaying 1–7 adherent cells). Columns, mean; error bars, s.d. (g) Representative micrographs displaying recovered primary MEFs at 3 hr post-arsenite stress. Scale bar, 50 μm. Statistics were calculated by Student's t-test, n.s., not significant; *p<0.05; ***p<0.0005. Data are representative of three independent double blind experiments. (h) Proportion of primary MEFs with eIF3[+] stress granules after 1 hr of 1 mM arsenite treatment comparing wild-type and *sanroque* MEFs recovering in complete DMEM media. Data are representative of two independent experiments (n >30 per time point, with each n replicate representing a single field of view displaying 1–6 adherent cells). Error bars, s.d. Statistics were calculated by Student's t-test, n.s., not significant.

The following figure supplements are available for Figure 6:

**Figure supplement 1.** Stress granule sequestration of RAPTOR.

**Figure supplement 2.** AMPK controls stress granule formation and maintenance.

mild in naive CD44[low] T cells but accentuated in CD44[high] cells. Reflecting a role for ROQUIN RING activity during early development, abated mTOR activity was also observed in ROQUIN RING deleted primary MEFs by enhanced phosphorylation of RAPTOR Ser[792] (*Figure 6b*), a target residue for AMPK-mediated inhibition.

Stress granules are AMPK-dependent (*Hofmann et al., 2012*; *Mahboubi et al., 2015*) cytoplasmic compartments that sequester and inactivate mTORC1 during cellular stress (*Thedieck et al., 2013*; *Wippich et al., 2013*). In primary MEFs, owing to their large cytoplasm and prominent stress granules, we confirmed RAPTOR localization to eIF3$^+$ stress granules in a wild-type and *Roquin$^{ringless}$* background (*Figure 6—figure supplement 1a, b*). We also found that arsenite-induced stress granule formation was impeded by AMPK inhibition in MEFs treated with Compound C (*Figure 6—figure supplement 2a*). Therefore, we sought to determine if diminished mTOR signaling was associated with augmented stress granule formation or maintenance in ROQUIN RING-deficient cells. Analysis of arsenite-stressed primary MEFs by fluorescence microscopy revealed that loss of ROQUIN RING signaling did not alter stress granule induction (*Figure 6c–e*) but rather prolonged the rate of stress granule dissolution during stress recovery (*Figure 6f, g*). A similar delay in stress granule recovery was mirrored in primary MEFs in which AMPK activity was raised upon treatment with AICAR (*Figure 6—figure supplement 2b*). Conversely, in *sanroque* mutant primary MEFs expressing a ROQUIN variant incapable of regulating target mRNAs, we found that stress granule recovery post-arsenite treatment was comparable to wild-type MEFs (*Figure 6h*). Together, these data suggest that the selective Tfh cell defect in *Tringless* mice may be a result of a disrupted ROQUIN–AMPK signaling axis, otherwise important in relieving stress granule inhibition of mTOR. Furthermore, ROQUIN RING-mediated stress granule subversion of mTOR activity appears to be independent of the RNA repressive functions of the ROQUIN ROQ domain.

## mTOR is required for optimal Tfh cell formation

To determine if attenuated mTOR is associated with defective Tfh cell responses as observed in *Tringless* animals, we examined *chino* mice harboring a hypomorphic mutation (*chi* allele) in the *Frap1* gene resulting in an Ile$^{205}$Ser substitution within the HEAT repeat domain of the mTOR protein (*Figure 7a*), the region dedicated to binding RAPTOR (*Kim et al., 2002*). Unlike the *in utero* lethality observed in mice with complete mTOR deficiency (*Gangloff et al., 2004*; *Murakami et al., 2004*), *chino* is a viable strain that exhibits growth retardation, intact thymocyte development and output but reduced phosphorylation of ribosomal protein S6 in phorbol-12-myristate-13-acetate treated peripheral CD4$^+$ T cells (*Daley et al., 2013*). We confirmed suboptimal phosphorylation of mTOR targets 4EBP1 and S6K in *chino* peripheral CD4$^+$ T cells in response to physiological TCR activation with CD28 costimulation (*Figure 7b*). This was in contrast to significantly elevated FOXP3 expression. Thus, *chino*-mutant T cells represent a mild deficiency in mTOR signaling reminiscent of a partial loss-of-function in mTOR (*Zhang et al., 2011*) that exclusively affects extrathymic CD4$^+$ T cell differentiation as seen in conditional T-cell-deleted *Frap1* knockout mice (*Delgoffe et al., 2009*). We immunized *chino* mice with SRBC, assessed the GC response 5 days later by flow cytometry and found that Tfh cells were severely diminished compared to wild-type controls (*Figure 7c*). This corresponded with a reduction in the GC B cell response (*Figure 7d*).

To determine if mTOR acts within Tfh cells, we constructed and immunized 50:50 mixed wild-type:*chino* bone marrow chimeras with SRBC. At d7 post-immunization, we confirmed that the percentages of mTOR mutant PD1 $^{high}$CXCR5 $^{high}$ Tfh cells were impaired compared to their competing wild-type counterparts in the same mouse (*Figure 7e*). This was associated with reduced BCL6 expression intrinsic to mTOR mutant CD4$^+$ T cells (*Figure 7f*). Our data therefore demonstrates that Tfh cells depend on intact intracellular mTOR signaling and that the *chino* Tfh-intrinsic phenotype closely mimics a defective ROQUIN RING deleted Tfh response. We therefore conclude that not only does mTOR act in the same polarity as the ROQUIN RING domain during Tfh cell development, but since mTOR signaling is a *bone fide* target of AMPK-directed inhibition (*Gwinn et al., 2008*; *Inoki et al., 2003*) in CD4$^+$ T cells (*Zheng et al., 2009*), it is most likely that mTOR represents a molecular pathway between the ROQUIN–AMPK axis and the control of Tfh responses (*Figure 7—figure supplement 1*).

## Discussion

The critical signaling requirements specific to programming Tfh cell differentiation have been under intense investigation in the past decade (*King and Sprent, 2012*; *Rolf et al., 2010*). ROQUIN, acting through its ROQ and C3H domains, has previously been identified as a potent post-transcriptional repressor of CD4$^+$ Tfh cells (*Glasmacher et al., 2010*; *Pratama et al., 2013*; *Vinuesa et al., 2005*;

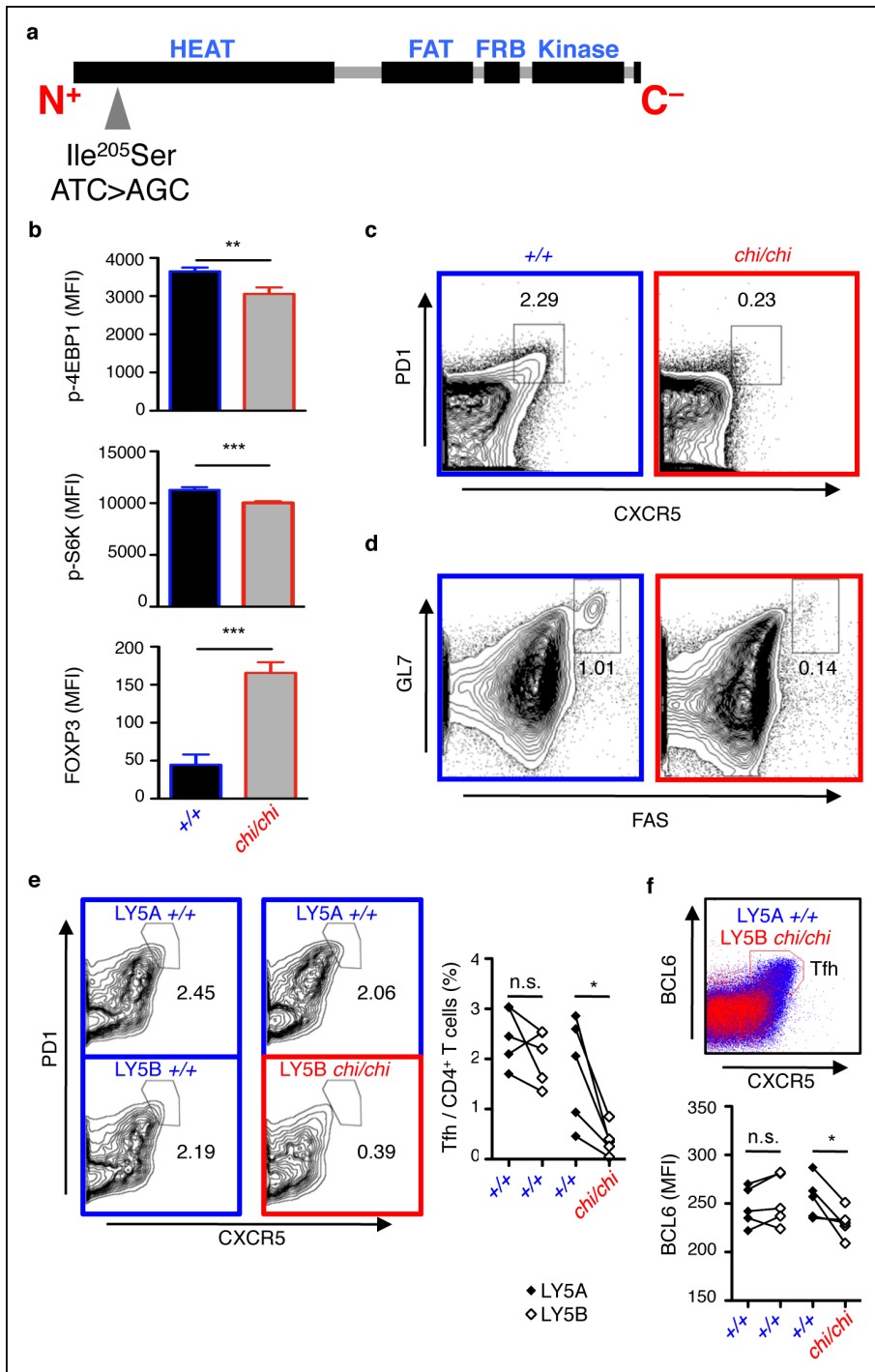

**Figure 7.** mTOR signaling is required for optimal Tfh cell formation. (a) *chino* mutation causes a I[205]S substitution in the mTOR protein. (b) Flow cytometric measurements of intracellular phospho-4EBP1 Thr[37/46], phospho-S6K Thr[389], and FOXP3 in CD4[+] T cells stimulated with anti-CD3 and -CD28 for 30 min (n = 4–6). MFI, mean fluorescence intensity; column, group mean, error bars, s.d. (c, d) *chino* mutants were immunized with sheep red blood cells (SRBC) and taken down 5 days later to analyze the proportion of PD1[high]CXCR5[high] Tfh cells from CD4[+] T cells (c), and the proportion of GL7[high]FAS[high] GC B cells from B220[+] B cells (d) in the spleen. Data are representative of three independent experiments. Statistics were calculated by Student's t-test, \*\*$p<0.005$; \*\*\*$p<0.0005$. (e, f) Flow cytometric analysis 50:50 mixed LY5A wild-type:LY5B *chino* bone marrow chimeras d7 post-SRBC immunization showing the proportion of PD1[high]CXCR5[high] Tfh cells (e), and expression of intracellular BCL6 (f) from the LY5A and LY5B CD4[+]B220[−] T cells. Linked dot symbols, congenic cells from same animal; MFI, mean fluorescence intensity. Data are representative of two independent experiments. Statistics were calculated by Paired Student's t-test between congenically marked cells of the same animal, \*$p<0.05$.

*Figure 7. continued on next page*

*Figure 7. Continued*

The following figure supplements are available for Figure 7:

**Figure supplement 1.** Schematic representation of ROQUIN signaling in Tfh cell ontogeny Secondary structure of the ROQUIN protein depicting the molecular function of its distinct domains in Tfh cells.

*Vogel et al., 2013*; *Yu et al., 2007*; *Lee et al., 2012*) but has also been shown to act similarly in limiting Th1, Th17 cells and CD8[+] effector T cells (*Bertossi et al., 2011*; *Jeltsch et al., 2014*; *Chang et al., 2012*; *Lee et al., 2012*). In the present study, we highlight for the first time the cellular function of the amino terminal ROQUIN RING finger and unexpectedly its importance as a positive immunomodulator of peripheral Tfh cells exclusively. In response to SRBC immunization and also to acute LCMV primary infection, mice lacking the ROQUIN RING domain in T cells failed to optimally form mature Tfh cells that could support a robust GC response. Interestingly, Th1, Th2, Th17, and Treg responses were comparable to wild-type controls *in vivo* and *in vitro*, depicting a functional uncoupling between RING signaling and ROQ-C3H activity in ROQUIN. This is consistent with our previous observations of ROQUIN RING-deficient T cells showing minimally altered expression of ICOS, a target of post-transcriptional repression (*Pratama et al., 2013*). We show at the molecular level, that the RING domain of ROQUIN is required to attenuate AMPK signals. In support of this finding, adenosine metabolism has previously been linked to T-dependent antibody responses in mice with observations that antigen-specific Tfh cells displayed constitutively high surface expression of CD73, an ecto-enzyme that catabolizes extracellular AMP into adenosine (*Iyer et al., 2013*; *Conter et al., 2014*). Taken together, it is possible that Tfh cells utilize a purinergic autocrine signaling pathway similarly suggested in Treg cells (*Deaglio et al., 2007*; *Sitkovsky, 2009*), whereby CD73-generated adenosine external to Tfh cells is imported through nucleoside transporters for reversion back into cytoplasmic AMP by adenosine kinase, before ROQUIN RING-regulated activation of AMPK. Furthermore, since AMPK is an inhibitor of glycolysis and cellular growth (*Hardie et al., 2012*; *Mihaylova and Shaw, 2011*), its activity in Tfh cells could facilitate BCL6 function, especially in transcriptionally dampening CD4[+] T cell glycolysis (*Oestreich et al., 2014*) which is otherwise important for cell growth (*Jones and Thompson, 2007*). This would form the basis for why Tfh cell numbers are so tightly contained throughout a GC response, acting as a critical limiting factor for controlling the magnitude and clonal diversity of GC reactions (*Schwickert et al., 2011*; *Victora and Nussenzweig, 2012*; *Rolf et al., 2010*). At first glance, it may seem conflicting that ROQUIN RING deficiency results in unrestrained AMPK leading to crippled BCL6 expression and Tfh cell hypocellularity. However, intact ROQUIN RING signaling may be advantageous, if not critical in Tfh responses, possibly acting to secure an intricate maximal threshold of AMPK activity that is key to maintaining narrow Tfh numbers for effective GC clonal selection while allowing ample, but not excessive, Tfh support to GCs.

We could not find any evidence that ROQUIN E3 ligase activity directly targets AMPK for proteosomal degradation. Instead, impaired autoubiquitination due to the absence of a functional RING domain suggests that ROQUIN E3 ligase activity can negatively regulate AMPK independent of the RNA regulatory functions of ROQUIN. Although AMPK β and γ subunits have been shown to localize to stress granules (*Mahboubi et al., 2015*), our results demonstrated an inability of the ectopically expressed regulatory subunits to localize with ROQUIN[+] stress granules in cells also overexpressing ROQUIN, hinting at the requirement of a different cofactor for their recruitment to stress granules. Given this and the present data demonstrating the ability of ROQUIN and AMPKα1 to bind each other and colocalize within stress granules together with previous observations of stress granule exclusion of RING-deficient ROQUIN (*Pratama et al., 2013*), we propose a model whereby ROQUIN may be repressing AMPK activity via ubiquitin-dependent sequestration of the AMPKα1 subunit within stress granules and thereby promoting repression of AMPK kinase activity. This stress granule-associated regulation of AMPK may not be exclusive to ROQUIN but could involve other binding partners such as G3BP1, which has been shown to localize with AMPKα2 in stress granules (*Mahboubi et al., 2015*). Interestingly, a direct interaction has also been observed between AMPKα and G3BP1 (*Behrends et al., 2010*), an integral component of stress granules that associates with ROQUIN (*Glasmacher et al., 2010*). Analogous to the T cell anergy-regulating RING-type E3 ubiquitin ligases GRAIL and CBL-B that undergo autoubiquitination (*Anandasabapathy et al., 2003*;

*Levkowitz et al., 1999*) as well as targeting various T cell signaling molecules for RING-mediated ubiquitination (*Nurieva et al., 2010*; *Su et al., 2006*; *2009*; *Lineberry et al., 2008*; *Fang et al., 2001*; *Fang and Liu, 2001*; *Jeon et al., 2004*), it is likely that ROQUIN also ubiquitinates additional proteins to coordinate Tfh cell immunity.

We found that ROQUIN RING deficiency results in intact thymic development and low phosphorylation levels of ribosomal S6 in a subset of peripheral CD4+ T cells, which together resembles closely *chino* mutant mice with impaired mTOR function (*Daley et al., 2013*). Our data points to overactive AMPK as the link between the loss of ROQUIN RING activity and reduced mTOR signaling. As AMPK activity has been shown to be transiently upregulated within 5–20 min of stress induction and to decline after the appearance of stress granules (*Mahboubi et al., 2015*), it is possible that the subcellular sequestration of AMPK within stress granules may represent a regulatory circuit breaker to interrupt the positive feed-forward loop that acts to shut down mTOR, mRNA translation and cell growth in response to cellular stress, AMPK induction and stress granules formation. Enhanced AMPK-mediated stress granule persistence leading to mTOR repression in the absence of ROQUIN RING signaling aligns well with a report of dampened TOR activity in eukaryotic cells during cellular stress by the transient shuttling of TOR into stress granules (*Takahara and Maeda, 2012*). In addition, *Wippich et al. (2013)* also found in HeLa cells having inactivated the stress granule inhibitory kinase DYRK3, that stress granule longevity was the key to prolonging mTOR inhibition.

Since *Rc3h1* appears to be ubiquitously expressed (*Vinuesa et al., 2005*), it is intriguing that the *Tringless* allele preferentially affects Tfh cells of the GC and no other CD4+ T cell lineage. We found in ROQUIN RING deleted CD4+ T cells that this may be a result, at least in part, of insufficient IL-21 cytokine production, which is otherwise required for optimal GC reactions and to a lesser degree, Tfh cell maintenance (*Zotos et al., 2010*; *Vogelzang et al., 2008*; *Linterman et al., 2010*). IL-21 deficiency would also explain why a normal Tfh cell response with diminished GC B cells in *Tringless* mice was detected early at d6 post-SRBC immunization (*Pratama et al., 2013*), but both mature GC Tfh and B cell populations were crippled at d8 in the current study. Within Tfh cells, ROQUIN RING activity may also be important for transducing stimuli downstream of a GC-specific receptor such as PD1, which is uniquely found most highly expressed on the surface of Tfh cells in humans and mice (*Yu and Vinuesa, 2010*; *Kamphorst and Ahmed, 2013*). In CD4+ T cells, ligation of PD1 couples mTOR signals (*Francisco et al., 2009*) and also restricts cellular glycolysis (*Patsoukis et al., 2015*; *Parry et al., 2005*) in a similar manner to AMPKα1 activity (*MacIver et al., 2011*; *Michalek et al., 2011*). It remains unclear how ROQUIN RING activity links to Tfh cell environmental stimuli, but there is evidence of ROQUIN phosphorylation by unknown kinases in human T cells (*Mayya et al., 2009*).

Previously, we showed that the *Rc3h1* 'sanroque' allele encodes a ROQUIN[M199R] mutant protein with a defective RNA-binding ROQ domain unable to repress ICOS. This, together with excessive IFNγ signaling causes aberrant accumulation of Tfh cells leading to unrestrained and pathogenic GC growth (*Lee et al., 2012*; *Yu et al., 2007*). The accumulation of Tfh cells in *sanroque* animals opposes the defective Tfh response of *Tringless* mice. We postulate that ROQUIN[133-1130] represents a complete loss of the AMPK-regulating functions with minimal disturbance to the RNA-regulating function. This may explain why the phenotype of mice with a combined deletion of the RING domains found in both ROQUIN and that of its closely related RC3H family member ROQUIN2 (*Pratama et al., 2013*) is less severe than the immune deregulation of *Roquin/Roquin2* double knockout mice (*Vogel et al., 2013*). We have previously shown that ROQUIN2 can compensate for the RNA-regulating function of ROQUIN, both repressing overlapping mRNA targets (*Pratama et al., 2013*). By contrast, the *sanroque* ROQUIN[M199R] mutated protein that can still localize to stress granules and bind RNA (*Athanasopoulos et al., 2010*) is likely to represent a recessive 'niche-filling' variant that selectively inactivates the normal mRNA-regulating function of ROQUIN but preserves its assembly into the mRNA decapping complex, preventing compensatory substitution by ROQUIN2 (*Pratama et al., 2013*). It is also likely that ROQUIN[M199R] protein found in *sanroque* T cells retains RING finger activity to negatively regulate AMPK and promote Tfh cell development, which is compounded by the increased stability of T cell mRNAs that exacerbate Tfh accumulation and trigger autoimmunity. Indeed in mice, T cell AMPK activity has been shown to play a protective role in autoimmune models of rheumatoid arthritis and multiple sclerosis (*Nath et al., 2009*; *Son et al., 2014*).

As a downstream target of AMPK metabolic signaling, mTOR is well known to orchestrate T cell effector differentiation and peripheral tolerance (*Chi, 2012*; *Araki et al., 2011*). As such, deregulation of mTOR can facilitate T-dependent autoimmune disorders like systemic lupus erythematosus (*Koga et al., 2014*; *Fernandez et al., 2006*; *Kato and Perl, 2014*) and multiple sclerosis (*Delgoffe et al., 2011*; *Esposito et al., 2010*). Although active mTOR signaling in Tfh cells has been documented (*Gigoux et al., 2014*), the specific role mTOR has in Tfh cell responses remains largely uncharacterized (*Araki et al., 2011*). Our data indicate that mTOR can act in concert with and downstream of ROQUIN RING signaling to support optimal Tfh cell formation. Susceptible to both ROQUIN-controlled AMPK repression and stress granule sequestration, mTOR regulation is thus integral to Tfh cell responses. Multiple studies are also in line with this model. Similar to our findings of impaired IL-21 synthesis in mTOR-attenuated *Tringless* CD4[+] T cells, a report on *Frap1* knockout T cells cultured *in vitro* also displayed reduced expression of IL-21 (*Delgoffe et al., 2009*). Moreover, low expression and secretion of IL-21 was observed in rapamycin-treated human CD4[+] T cells polarised toward the Tfh cell lineage *ex vivo* (*De Bruyne et al., 2015*). Also in mice with reduced mTOR, T-dependent B cell proliferation, isotype switching, GC formation and antigen-specific antibody responses were significantly crippled (*Zhang et al., 2011*; *Keating et al., 2013*). Additional *in vivo* studies, however, are required to dissect the downstream signals transduced by mTOR that orchestrate Tfh immune responses. Thorough investigation of these and similar metabolic pathways including AMPK-dependent cellular bioenergetics within Tfh cells and in other GC T cell subsets (*Ramiscal and Vinuesa, 2013*) is not only warranted but also may advance current strategies for vaccine design or reveal novel therapeutic interventions for antibody-mediated immune disorders.

## Materials and methods

### Mice

ROQUIN RING deleted mice (*Pratama, et al., 2013*) were generated by Ozgene, Australia; *loxP* sites that flanked or 'floxed' *Rc3h1* exon 2, encoding the START codon and RING motif (*Figure 1—figure supplement 1*), were inserted into C57BL/6 mouse embryonic stem (ES) cells via homologous recombination . Recombinant ES cell clones were implanted into C57BL/6 foster mothers. Heterozygote progeny were screened for germline transmission before crossing to *Rosa26:Flp1* mice to remove the *neo* cassette. Mice harboring the floxed *Rc3h1* allele (*Rc3h1[lox/+]*) were then crossed to *Rosa26:Cre* knock-in mice for one generation. Removal of Cre expression was then achieved by a C57BL/6 backcross yielding a germline deletion of *Rc3h1* exon 2. This strain was named *ringless* (*rin* allele). A conditional ROQUIN RING-deficient strain (called *Tringless*) was also generated by crossing *Rc3h1[lox/lox]* mice to *Lck:*Cre breeders to remove *Rc3h1* exon 2 specifically in T lymphocytes (*Trin* allele). Upon Cre-mediated excision of *Rc3h1* exon 2, rescue of in-frame protein synthesis at Met[133] is expected to produce an E3 ligase defective ROQUIN mutant. *Rosa26:Cre* and *Lck:Cre* mice were maintained on a C57BL/6 background with one copy of the Cre transgene and provided by Ozgene, Australia. ENU-derived *chino* mutants were previously characterized (*Daley et al., 2013*). To generate mixed bone marrow chimeric mice, recipient *Rag1[-/-]* mice were sublethally irradiated and reconstituted i.v. with 2 x 10[6] bone marrow hematopoietic stem cells.

Animal experiments were approved by the Animal Experimentation Ethics Committee of the Australian National University (Protocols J. IG.71.08, A2012/05 and A2012/53) and the McGill University Ethics Committee (Protocol 7259). Mice were maintained in a specific germ-free environment. Where indicated, 8 to 12 wo mice were immunized i.p. with 2 x 10[9] SRBC to generate a T-dependent GC response or i.p. with 2 x 10[5] PFU of LCMV Armstrong.

### Molecular reagents

The following antibodies were used in Western blots, immunoprecipitation assays and fluorescence microscopy: rabbit anti-phospho-ACC Ser[79] (Cat. 3661, Cell Signaling), rabbit phospho-RAPTOR Ser[792] (Cat. 2083, Cell Signaling), rabbit anti- β-ACTIN (13E5, Cell Signaling), rabbit anti-AMPKα (Cat. ab32047, Abcam, UK), goat anti-eIF3 (N-20, Santa Cruz), mouse anti-GFP (7.1 and 13.1, Roche), rabbit anti-GFP (Cat. ab290, Abcam), mouse anti-HA (HA-7, Sigma-Aldrich), rabbit anti-HA (H6908, Sigma-Aldrich), rabbit anti-RAPTOR (24C12, Cell Signaling), rabbit anti-ROQUIN (Cat. A300-514A, Bethyl Laboratories), mouse anti-UBIQUITIN (P4D1, Cell Signaling), mouse anti-V5 (V5-10, Sigma-

Aldrich), rabbit anti-V5 (Cat. V8137, Sigma- Aldrich), mouse anti-rabbit IgG light chain (211-032-171, Jackson ImmunoResearch), and goat anti-mouse IgG light chain (155-035-174, Jackson ImmunoResearch). AICAR (Calbiochem) and Compound C (Calbiochem) were used according to manufacturers' recommendations at indicated concentrations. N-terminal V5 tagged full length ROQUIN and ROQUIN[133-1130] constructs have been previously described (*Pratama et al., 2013*). C-terminal GFP fused ROQUIN and ROQUIN[C14A] constructs have previously been described (*Athanasopoulos et al., 2010*). GFP tagged constructs of AMPK subunits were obtained from Origene. HEK293T and EL4 cells were obtained from the ATCC and perpetuated in-house. Primary MEFs were harvested from E14 fetuses of *rin/+* or *san/+* pregnant females that were paired with *rin/+* or *san/+* males, respectively, as part of a timed mating.

## T cell stimulation, flow cytometry, and immunofluorescence

Where indicated *in vitro* stimulation of T cells was performed using anti-CD3 and anti-CD28 dual coated Dynabeads (Invitrogen) or for cytokine accumulation, phorbol myristate acetate (Sigma-Aldrich), and ionomycin (Sigma-Aldrich) was used with GolgiStop (BD Biosciences) in RPMI 1640 medium (Invitrogen) supplemented with 2 mM l-glutamine (Invitrogen), 100 U penicillin-streptomycin (Invitrogen), 0.1 mM non-essential amino acids (Invitrogen), 100 mM HEPES (Sigma-Aldrich), 0.0055 mM 2-mercaptoethanol, and 10% FCS. 20 ng/mL IL-2 (R&D Systems), 100 ng/mL IL-4 (Miltenyi Biotec), 100 ng/mL IL-6 (Peprotech), 20 ng/mL IL-12 (Miltenyi Biotec), 1 ng/mL TGFβ (R&D Systems), along with 1µg/mL of Biolegend antibodies anti-IL-4, anti-IFNγ and/or anti-IL-12 were used for *in vitro* polarization of (IL-2) Th0, (IL-2, IL-12, and anti-IL-4) Th1, (IL-2, IL-4, anti-IFNγ, and anti-IL-12) Th2, (IL-2, IL-6, TGFβ, anti-IL-4, and anti-IFNγ) Th17, and (IL-2 and TGFβ) iTreg cultures. To stain surface markers, cells were washed and stained in ice-cold staining buffer (2% FCS, 0.1% NaN$_3$ in PBS). eBioscience FOXP3 Staining Buffer Set was used for flow cytometric detection of intracellular proteins. Data were acquired by a LSRII Flow Cytometer using FACSDiva software. MEFs and HEK293T cells were prepared for fluorescence microscopy as previously described (*Athanasopoulos et al., 2010*). Images were collected using an Olympus IX71 microscope with DP Controller software (Olympus).

## AMPK kinase assay in CD4[+] T cells

CD4[+] T cells were isolated from *floxed* wild-type or *Tringless* mice by MACS Microbead separation (Miltenyi Biotec). AMPK activity was measured from AMPK complexes immunoprecipitated from cell lysates using anti-AMPKα antibody (Abcam) as previously described (*Chen et al., 2003*). Detection of ACC Ser[79] phosphorylation levels was also used to measure AMPK allosteric activity.

## Immunoprecipitation and Western blotting

Whole-cell lysates were prepared using TNE lysis buffer (1% NP40, 150 mM NaCl, 20 mM Tris-base, 1 mM EDTA and Roche cOmplete EDTA-free protease inhibitory cocktail tablets all dissolved in water). PhosSTOP (Roche) was added to the TNE mix for the detection of phospho-residues. To immunoprecipitate proteins, antibody was added to pre-cleared lysates and mixed with Protein G Sepharose 4 Fast Flow (GE Healthcare) for 12 hr then washed. For western blotting, lysates were separated by SDS-PAGE, transferred to nitrocellulose membrane, blocked in 5% BSA Tris-buffered saline containing 0.05% Tween-20, probed with primary antibodies and detected with horseradish peroxidase-conjugated anti-rabbit or anti-mouse secondary antibodies.

## Proximity ligation assays

MEFs were seeded on coverslips and prepared as described previously (*Srivastava et al., 2015*). To induce cellular stress, 1 mM arsenite was added to cultures for 1 hr. Stains with primary antibodies were carried out using optimized conditions overnight at 4°C in a humid chamber. The primary antibodies were goat anit-AMPKα1, clone C20 (Santa Cruz) at 1:75 ; with either rabbit anti-ROQUIN at 1:75 (Cat. NB100-655, Novus Biologicals) or rabbit anti-GFP 1:1000 (Cat. ab6556, Abcam, UK). Images were taken on a Leica SP5 confocal microscope with a pin hole of 67.9 µm and an APO CS 1.25 UV x40 oil objective. Higher magnification images presented in *Figure 3c* were taken on a Leica SP5 confocal microscope with a pin-hole of 95.5 µm and an HCxPL APO lambda blue x63/1.4 oil objective.

## *In vitro* autoubiquitination assays

Mouse UBCH5A (E2) was expressed as a GST-fusion protein and purified using standard protocols. ROQUIN constructs were also expressed as GST-fusion proteins using standard procedures except that 0.1 mM Zn-acetate was added to the growth media and all purification buffers. Human E1 (His$_6$ tagged) was purchased from Biomol International. Bovine UBIQUITIN was purchased from Sigma-Aldrich. Ubiquitination assays were performed in 20 ml in 20 mM Tris-HCl, 50 mM NaCl, 2 mM MgCl$_2$, 1 mM ATP, 0.1 mM DTT at 25°C. Reactions were stopped by the addition of 2x SDS PAGE loading buffer and heating at 95°C for 5 min and analyzed by SDS-PAGE and Coomassie Blue staining. Typically reactions contained 0.1 mM E1, 10 mM E2, 50 mM UBIQUITIN, and 0.5 mg/mL ROQUIN peptide.

## *In silico* analysis

Statistics were calculated using Prism 5.0a software (GraphPad). Stress granule morphology and PLAs were assessed by ImageJ 1.46r software (NIH).

# Acknowledgements

We thank the Biomolecular Resource Facility (BRF), the Microscopy and Cytometry Resource Facility (MCRF), and the animal services and genotyping teams at the Australian Phenomics Facility (APF) for animal support. CGV is an Elizabeth Blackburn Fellow of the NHMRC. MAF is a Senior Principal Research Fellow of the NHMRC. CCG is an Australian Fellow of the NHMRC. RSLY is a Career Development Fellow of the NHMRC.

# Additional information

## Funding

| Funder | Grant reference number | Author |
|---|---|---|
| National Health and Medical Research Council | NHMRC Elizabeth Blackburn Fellowship | Carola G Vinuesa |
| National Health and Medical Research Council | NHMRC Senior Principle Research Fellowship | Mark A Febbraio |
| National Health and Medical Research Council | NHMRC Career Development Fellowship | Robert S Lee-Young |
| National Health and Medical Research Council | NHMRC Project Grant APP1061580 | Carola G Vinuesa |
| National Health and Medical Research Council | NHMRC Program Grant APP1016953 | Christopher C Goodnow Carola G Vinuesa |

The funders had no role in study design, data collection and interpretation, or the decision to submit the work for publication.

## Author contributions

RRR, CGV, VA, Conception and design, Acquisition of data, Analysis and interpretation of data, Drafting or revising the article, Contributed unpublished essential data or reagents; IAP, RSLY, JJB, Acquisition of data, Analysis and interpretation of data, Drafting or revising the article; JB, RGJ, MAF, Acquisition of data, Analysis and interpretation of data, Drafting or revising the article, Contributed unpublished essential data or reagents; AP, JM, NH, JYC, PFN, JIE, NJK, RAS, Acquisition of data, Analysis and interpretation of data, Contributed unpublished essential data or reagents; CCG, Conception and design, Analysis and interpretation of data, Contributed unpublished essential data or reagents

## Ethics

Animal experimentation: Mouse studies were approved by the Animal Experimentation Ethics Committee of the Australian National University (Protocols J.IG.71.08, A2012/05 and A2012/53) and the McGill University Ethics Committee (Protocol 7259). Mice were maintained in a specific germ-free environment.

## Additional files

### Major datasets

The following previously published dataset was used:

| Author(s) | Year | Dataset title | Dataset ID and/or URL | Database, license, and accessibility information |
|-----------|------|---------------|------------------------|--------------------------------------------------|
| Kershaw NJ, Vinuesa CG, Babon JJ | 2015 | Crystal structure of N-terminus of Roquin | http://www.rcsb.org/pdb/explore/explore.do?structureId=4txa | Publicly available at the RCSB Protein Data Bank (Accession no: 4TXA)1) |

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
