## [Decision Letter]

Thank you for submitting your work entitled "Attenuation of AMPK signaling by ROQUIN promotes formation of T follicular helper cells" for peer review at *eLife*. Your submission has been favorably evaluated by Tadatsugu Taniguchi (Senior Editor) and three reviewers, one of whom is a member of our Board of Reviewing Editors. One of the three peer reviewers, Masato Kubo, has agreed to reveal his identity.

The reviewers have discussed the reviews with one another and the Reviewing Editor has drafted this decision to help you prepare a revised submission.

The post-transcriptional repressor ROQUIN has previously been demonstrated to have a critical role in the negative regulation of Tfh cells as *sanroque* mice with a dominant negative ROQUIN mutation or T cell-specific deletion of ROQUIN and its paralogue ROQUIN2 leads to unrestrained formation of Tfh. The authors here investigate the effect of specific depletion of the RING domain of ROQUIN and unexpectedly find that this domain has a positive role in Tfh formation in contrast to the overall negative role of ROQUIN. They describe that this loss of Tfh formation and function is due to loss of ROQUIN RING dependent antagonism of AMPK which in turn causes loss of mTOR function that, through largely unknown mechanisms, inhibits Tfh formation.

Major comments:

1) In Figure 7, Foxp3 MFI rose in the *chino* mutants. As the authors of this paper have previously demonstrated CD4^+^PD1^+^CXCR5^+^ cells are made up of a mixture of Foxp3- Tfh and FOXP3^+^ Tfr cells. A rise in the MFI of Foxp3 fails to determine if this is Foxp3 expression being induced by Tfh or, as seems more likely, a change in the proportion of Foxp3 expressing Tfr cells within the "Tfh" gate. Since Tfh and Tfr are distinct populations it would be better to more clearly divide the population into Foxp3- Tfh and FOXP3^+^ Tfr and showing this both by representative gating and including a graph of % Foxp3 within "Tfh" or similar method. It is important that the authors have demonstrated in Figure 1—figure supplement 2 that Thymic Treg formation and Treg levels in the periphery are not significantly altered in *Trin/Trin* mice. However this does not fully address the question of Tfr formation as we do not understand the role of *Trin* in Tfr. In much the same way that looking at total CD4 cells would not have revealed the effects on Tfh looking at total Tregs will not reveal any Tfr specific effects. Given the known role of mTOR inhibitors such as rapamycin in enhancing Treg numbers and the data in Figure 7, it is plausible that the Tfh/Tfr ratio may be altered. Addressing this point will both strengthen the conclusions on Tfh (by confirming what percentage of them are actually Tfh) and add impact by widening the results to include or exclude an effect on Tfr. This issue can be quickly and simply addressed by including some indication of the percentage of Foxp3 expression within the PD1^+^CXCR5^+^ population Tfh in Figure 1 and Figure 7. Presumably the data in Figure 7 is already available and simply requires reanalysis while Figure 1 may require a further experiment.

2) Are these mice the same *Roquin1 ringless* mice previously used in Pratama et al. Immunity 2013? If so this could be referenced more clearly in the Methods section as the previous paper also described these mice to some extent. Of note in Pratama et al. 2013 it was shown in Figure 4 that *Trin/Trin Lck:Cre* mice had reduced GC formation and a trend towards reduced Tfh formation following SRBC vaccination. That is essentially the same experiment as seen here in Figure 1—figure supplement 2. It would be harsh to suggest that this affects novelty, since the previous paper did not focus on this issue and the current paper goes into far more detail. Nonetheless the authors should more clearly clarify this work in its relation to previously published work by noting this previous finding.

3) Figure 1 and Figure 2 address the ROQUIN-mediated mTOR attenuation cause specific defect in Tfh during LCMV infection, and Figure 7 indicated that hypomorphic mutation mice of the *Frap1* gene also had the same phenotype. However, previous reports have suggested that mTOR signals are also required for Th1, Th17, and Th2 differentiation (Kopf et al. 2007 Int. Immunopharmacol.; Delgoffe et al. 2009 Immunology; Delgoffe et al. 2011 Nat. Imunol.). I understand that different lineages have different requirements in mTOR signal. To strengthen their hypothesis, the authors need a more mechanistic explanation why the defect of RING domain loss appeared only in the Tfh program.

4) The manuscript does very well at describing how the ROQUIN-AMPK-mTOR axis controls the expression of several key genes involved in the Tfh program, CXCR5 and *Bcl6*. It raises the question of how the ROQUIN-AMPK-mTOR axis controls the expression of CXCR5 and *Bcl6* and what is a target of this pathway. It is recommended that the authors show transcriptome data of TFH cells in the ROQUIN RING loss mice and the hypomorphic mutant mice of the *Frap1* gene and compare.

5) The FACS profile indicating GC-B in the hypomorphic mutant mice of the *Frap1* gene is clearly different from that in the ROQUIN RING loss mice (Figure 1). We hope you can show the impact on humoral responses, because TFH and GC-B cell development is not convincing enough to prove the impact in humoral responses. It is recommended they show the antibody responses in the response to LCMV.

6) The finding that the effects seen are independent of *Roquin's* RNA binding activity and its recruitment of the CCR4-NOT complex to RNA targets is particularly interesting. Nonetheless, some aspects of the data provided are not entirely compelling. Both proteins, ROQUIN and AMPKα1, colocalize but the association as measured by co-immunoprecipitation, seems quite inefficient. The authors should consider being more circumspect in this conclusion and allow for the possibility that the interactions are indirect. Also, some of the lanes shown in Figure 4 are so overexposed that it is not clear what they add. Other blots are severely cropped – such blots are less convincing. In addition, it would seem that showing the control of a mutant with abrogation of the RNA binding could be useful in the stress granule assays.

7) The main data demonstrating an alteration in mTOR signaling in *Trin* mice is presented in Figure 6. Showing the primary data for a selected experiment (i.e. histograms of pS6) would be more convincing. More evidence supporting this conclusion would strengthen the authors' argument.

8) The authors suggest that this effect seen is specific to Tfh cells. This may be the case; however, their investigation into alternate T cell subsets is limited. *In vitro* Th1, Th2, Th17, and iTreg differentiation would be the minimum exploration using the *Trin* mice, and it would be additionally helpful to explore antigen specific Th1 differentiation in LCMV (Figure 2, add tetramer+ Th1 cells). Additionally, exploration of these cell types in the *chi/chi* model would be helpful to confirm that this is truly specific to Tfh cell differentiation. An additional angle that is lacking is the role of *Roquin2* in compensating for *Roquin1*. Previous work from this group has shown that *Roquin2* can compensate for *Roquin1*. Do doubly deficient *Roquin1/Roquin2* mice have a more severe defect in Tfh differentiation, or does it extend beyond Tfh cells? Are there circumstances in which compensation is more relevant? *Roquin* appears to be broadly expressed and the formation of stress granules is also not cell-type specific (and presumably will require AMPK for efficient resolving/modulation of TOR). Better explanation of why Tfh cells are selectively affected is in order.

---

## [Author Response]

*Major comments:1) In Figure 7, Foxp3 MFI rose in the chino mutants. As the authors of this paper have previously demonstrated CD4^+^PD1^+^CXCR5^+^cells are made up of a mixture of Foxp3- Tfh and FOXP3^+^ Tfr cells. A rise in the MFI of Foxp3 fails to determine if this is Foxp3 expression being induced by Tfh or, as seems more likely, a change in the proportion of Foxp3 expressing Tfr cells within the "Tfh" gate. Since Tfh and Tfr are distinct populations it would be better to more clearly divide the population into Foxp3- Tfh and FOXP3^+^ Tfrand showing this both by representative gating and including a graph of% Foxp3 within "Tfh" or similar method. It is important that the authors have demonstrated in Figure 1—figure supplement 2 that Thymic Treg formation and Treg levels in the periphery are not significantly altered in Trin/Trin mice. However this does not fully address the question of Tfr formation as we do not understand the role of Trin in Tfr. In much the same way that looking at total CD4 cells would not have revealed the effects on Tfh looking at total Tregs will not reveal any Tfr specific effects. Given the known role of mTOR inhibitors such as rapamycin in enhancing Treg numbers and the data in Figure 7, it is plausible that the Tfh/Tfr ratio may be altered. Addressing this point will both strengthen the conclusions on Tfh (by confirming what percentage of them are actually Tfh) and add impact by widening the results to include or exclude an effect on Tfr. This issue can be quickly and simply addressed by including some indication of the percentage of Foxp3 expression within the PD1^+^CXCR5^+^ population Tfh in Figure 1 and 7. Presumably the data in Figure 7 is already available and simply requires reanalysis while Figure 1 may require a further experiment.*

We and others (Ramiscal and Vinuesa, 2013 *Immunological Reviews*; Sage et al. 2013 *Nature Immunology*; Ding et al. 2014 *Arthritis and Rheumatology*) have proposed that compared to the conventional CD4^+^PD1^+^CXCR5^+^ total “Tfh gate”, the evaluation of distinct subsets within this total Tfh population, especially the ratio between FOXP3^-^ effector Tfh cells and their FOXP3 expressing suppressive Tfr counterparts may serve as an important tool for projecting the magnitude and robustness of T-dependent antibody responses and GC reactions. Considering that Tfr cells are a relatively new T cell subset, incompletely understood and their thymic origin differing from the peripheral development of effector Tfh cells, we are cautious in explicitly associating Tfr differentiation with ROQUIN RING signaling since arguing for this case would extend beyond the focus of the present study and would require further extensive investigation. Nonetheless, *Tringless* show an augmented population of FOXP3^+^ Tfr cells at the expense of effector Tfh cells (new Figure 2), both subsets likely competing for a shared GC niche. This trend is also reflected in immunized *chino* mice (data not published).

*2) Are these mice the same Roquin1 ringless mice previously used in Pratama* et al. *Immunity 2013? If so this could be referenced more clearly in the Methods section as the previous paper also described these mice to some extent. Of note in Pratama* et al*. 2013 it was shown that Trin/Trin Lck:Cre mice had reduced GC formation and a trend towards reduced Tfh formation following SRBC vaccination. That is essentially the same experiment as seen here in Figure 1—figure supplement 2. It would be harsh to suggest that this affects novelty, since the previous paper did not focus on this issue and the current paper goes into far more detail. Nonetheless the authors should more clearly clarify this work in its relation to previously published work by noting this previous finding.*

The *Lck:Cre ringless* mice in our 2013 *Immunity* report are indeed the same as the *Rc3h1 Tringless* mice presented in the current study. The former case depicts early Tfh percentages at d6 post-SRBC immunization. In contrast, the present study provides a d8 FACS analysis of mature Tfh responses to SRBC immunization in *Tringless*. This d8 Tfh response is reconciled with the findings in Figure 1 and Figure 2 showing Tfh responses to LCMV d10 post-infection. Based on our previous examination on the influence of IL-21 cytokine in SRBC-responsive Tfh responses (Linterman et al., 2010 *Journal of Experimental Medicine*), it is evident that IL-21 deficiency may begin to destabilize Tfh formation and GC reactions at d8 post-immunization but not on d6. We have included additional data (new Figure 2) demonstrating that *Tringless* CD4^+^ T cells produce less IL-21, and this may, at least in part, provide some insight into the different Tfh percentages detected in *Tringless* mice at d6 versus d8 post-SRBC injection. As advised by the reviewers, we have endeavored to more clearly relate the present study to our previously published Pratama et al. (2013, *Immunity*) findings (citation in Materials and methods, Mice section and new text in the Introduction, last paragraph).

*3) Figure 1 and Figure 2 address the ROQUIN-mediated mTOR attenuation cause specific defect in Tfh during LCMV infection, and Figure 7 indicated that hypomorphic mutation mice of the Frap1 gene also had the same phenotype. However, previous reports have suggested that mTOR signals are also required for Th1, Th17, and Th2 differentiation (Kopf et al. 2007 Int. Immunopharmacol.; Delgoffe et al. 2009 Immunology; Delgoffe et al. 2011 Nat. Imunol.). I understand that different lineages have different requirements in mTOR signal. To strengthen their hypothesis, the authors need a more mechanistic explanation why the defect of RING domain loss appeared only in the Tfh program.*

We have now proposed a more in-depth model for why ROQUIN RING activity is specifically affecting Tfh cell responses (Discussion, fourth paragraph). We postulate that reduced IL-21 production and maybe deregulated PD1 signal transduction may be factors in the Tfh-specific phenotype of immunized *Tringless* mice.

*4) The manuscript does very well at describing how the ROQUIN-AMPK-mTOR axis controls the expression of several key genes involved in the Tfh program, CXCR5 and Bcl6. It raises the question of how the ROQUIN-AMPK-mTOR axis controls the expression of CXCR5 and Bcl6 and what is a target of this pathway. It is recommended that the authors show transcriptome data of TFH cells in the ROQUIN RING loss mice and the hypomorphic mutant mice of the Frap1 gene and compare.*

As advised by the reviewers, we attempted to FACS sort Tfh cells from both SRBC-immunized *Tringless* and *chino* mice but were unable to obtain enough Tfh cells for quality RNA input for transcriptome analysis. In our hands, it is not feasible to sufficiently sort Tfh cells from GC-immunodeficient mice such as these. However, we have identified a deficiency in T cell production of the GC-specific cytokine IL-21 in immunized *Tringless* mice (new Figure 2). IL-21 expression is known to be mTOR-dependent *in vivo* and *in vitro* as noted by published reports, which are cited in the Discussion section (last paragraph).

*5) The FACS profile indicating GC-B in the hypomorphic mutant mice of the Frap1 gene is clearly different from that in the ROQUIN RING loss mice (Figure 1). We hope you can show the impact on humoral responses, because TFH and GC-B cell development is not convincing enough to prove the impact in humoral responses. It is recommended they show the antibody responses in the response to LCMV.*

We agree with the reviewers’ observation and note the distinct GC B cell responses in the two mouse models examined in our study. One explanation for the varied GC reactions in *Tringless* versus the *chino* strain is that in *Tringless* mice, all B cells are ROQUIN RING sufficient but dependent on ROQUIN RING-deleted T cell help, whereas in *chino* animals, defective mTOR signaling is intrinsic not only to T cells but also B cells since all germline cells are mutated. Zhang et al. (2014) has previously outlined an important B cell-intrinsic role for mTOR in GC reactions in response to various T-dependent antigens. Nonetheless, as suggested we have performed an ELISA assay to measure anti-LCMV IgG and as expected found no significant differences in antibody titres between *Tringless* and floxed wild-type animals at d10 post-viral infection (Figure 8). Similar findings have also been reported in Tfh-deficient CXCR5 KO (Fahey et al. 2011, *Journal of Experimental Medicine*) and SAP KO (Crotty et al. 2003, *Nature*) mice infected with LCMV and examined 10 d later. This is because early antibody responses to primary infection are predominantly derived from short-lived extrafollicular responses and less dependent on GC reactions which tend to play a critical role in the generation of affinity-matured long-term antibody responses and most importantly in seeding memory B cells for secondary recall antibody production. As such, with multiple older studies demonstrating Tfh-independent early anti-viral antibody responses, we believe addition of our data showing *Tringless* early IgG responses being intact at d10 post-infection will not add significant novelty nor dimension to our study.

Author response image 1.*Tringless* mice inoculated with LCMV and sera collected d10 post-infection to measure the anti-LCMV specific IgG response. n.s., not significant.**DOI:**
http://dx.doi.org/10.7554/eLife.08698.017

*6) The finding that the effects seen are independent of Roquin's RNA binding activity and its recruitment of the CCR4-NOT complex to RNA targets is particularly interesting. Nonetheless, some aspects of the data provided are not entirely compelling. Both proteins, Roquin and AMPKα1, colocalize but the association as measured by co-immunoprecipitation, seems quite inefficient. The authors should consider being more circumspect in this conclusion and allow for the possibility that the interactions are indirect. Also, some of the lanes shown in Figure 4 are so overexposed that it is not clear what they add. Other blots are severely cropped – such blots are less convincing. In addition, it would seem that showing the control of a mutant with abrogation of the RNA binding could be useful in the stress granule assays.*

We agree with the reviewers’ comments that the immunoprecipitation assays do not exclude the possibility that the ROQUIN-AMPKα1 interaction is indirect. We therefore performed an *in situ* proximity ligation assay (PLA), which detects interacting proteins that are in close proximity, within 30-40 nm of each other, and allows for a quantitative analysis of these interactions. Our PLA results indicate that ROQUIN and AMPKα1 endogenous proteins in primary C57BL/6 mouse embryonic fibroblasts (MEFs) are close binding partners (new Figure 3). In Figure 4, we agree that the over-expressed lanes may not significantly add to the data as intended and thus they have been removed (new Figure 4). We have also expanded the size of cropped western blot images to show more length in the lanes. Additionally, as suggested by the reviewers we performed stress granule recovery assays of *sanroque* MEFs which express the ROQUIN^M199R^ mutant protein with a dysfunctional ROQ domain incapable of repressing mRNA. We found that *sanroque* MEFs recovered as efficiently from arsenite-induced cellular stress as wild-type MEFs (new Figure 6). This is in contrast to the *ringless* MEFs which display a significant lag during stress granule recovery. These results further highlight a ROQUIN RING functional divergence from ROQ-mediated RNA metabolism.

*7) The main data demonstrating an alteration in mTOR signaling in Trin mice is presented in Figure 6. Showing the primary data for a selected experiment (i.e. histograms of pS6) would be more convincing. More evidence supporting this conclusion would strengthen the authors' argument.*

As requested by the reviewers, we have now replaced our line graph of p-S6 with its histogram and have added extra data to show p-S6 data in Tnaive CD44^low^ cells as well as antigen-experienced CD44^high^ cells (new Figure 6). Furthermore, we have also included data demonstrating that mTOR-dependent IL-21 cytokine production is impaired in immunized *Tringless* mice (new Figure 2).

*8) The authors suggest that this effect seen is specific to Tfh cells. This may be the case; however, their investigation into alternate T cell subsets is limited. In vitro Th1, Th2, Th17, and iTreg differentiation would be the minimum exploration using the Trin mice, and it would be additionally helpful to explore antigen specific Th1 differentiation in LCMV (Figure 2, add tetramer+ Th1 cells). Additionally, exploration of these cell types in the chi/chi model would be helpful to confirm that this is truly specific to Tfh cell differentiation. An additional angle that is lacking is the role of Roquin2 in compensating for Roquin1. Previous work from this group has shown that Roquin2 can compensate for Roquin1. Do doubly deficient Roquin1/Roquin2 mice have a more severe defect in Tfh differentiation, or does it extend beyond Tfh cells? Are there circumstances in which compensation is more relevant? Roquin appears to be broadly expressed and the formation of stress granules is also not cell-type specific (and presumably will require AMPK for efficient resolving/modulation of TOR). Better explanation of why Tfh cells are selectively affected is in order.*

We have examined T cell cultures from sorted CD4^+^ naive T cells to look at the influence of ROQUIN RING deficiency in the differentiation of Th1, Th2, Th17 and iTreg cells *in vitro* (new Figure 1—figure supplement 2). Our new data demonstrate that the expression of nuclear factors required for non-follicular effector T cell development in the periphery, namely TBET, GATA3, RORγT and FOXP3 in activated CD4^+^ T cell cultures are comparable to wild-type cells. This is consistent with our *in vivo* findings in mice immunized with SRBC or infected with LCMV. It is not likely that *chino* mice display a Tfh-specific defect as observed in *Tringless* animals since *chino* T cells represent a ROQUIN-indepedent mTOR defect and display a widespread inability to generate and/or sustain a CD44^high^ effector and memory T cell compartment (Daley et al. Elife, 2013). It is possible that ROQUIN RING signaling acts as a signal transducer of a Tfh-specific signal such as PD1, FR4 or CD73 found highly expressed on the surface of Tfh cells, to control mTOR signals.

Regarding the question of whether doubly deficient *Roquin1/Roquin2* mice have a more severe defect in Tfh differentiation, we showed in Pratama et al. *Immunity* (2013) that the Tfh defect observed in *Roquin1 ringless* mice is not seen in *Roquin2 ringless* and that double ROQUIN1/2 RING deficiency in fact corrects the *Roquin1 ringless* Tfh defect. These results suggest that ROQUIN2 does not target AMPK in Tfh cell responses, and additional targets may play opposing roles on Tfh cells, but elucidation of the relevant ROQUIN2 actions to explain this effect we believe are beyond the scope of the current manuscript.

As advised, we have included the tetramer^+^ Th1 cell response of LCMV-infected *Tringless* mice (Figure 2, Figure 2) showing no abnormalities when compared to wild-type floxed controls. These observations of virus-specific Th1 cells are in line with intact total Th1 responses (Figure 1).